# Femtosecond proton transfer in urea solutions probed by X-ray spectroscopy

Zhong Yin[1,6,8 ✉], Yi-Ping Chang[2,7,8], Tadas Balčiūnas[1,2,8], Yashoj Shakya[3,4,8], Aleksa Djorović[2], Geoffrey Gaulier[2], Giuseppe Fazio[1], Robin Santra[3,4,5], Ludger Inhester[3,5 ✉], Jean-Pierre Wolf[2 ✉] & Hans Jakob Wörner[1 ✉]

Proton transfer is one of the most fundamental events in aqueous-phase chemistry and an emblematic case of coupled ultrafast electronic and structural dynamics[1,2]. Disentangling electronic and nuclear dynamics on the femtosecond timescales remains a formidable challenge, especially in the liquid phase, the natural environment of biochemical processes. Here we exploit the unique features of table-top water-window X-ray absorption spectroscopy[3–6] to reveal femtosecond proton-transfer dynamics in ionized urea dimers in aqueous solution. Harnessing the element specificity and the site selectivity of X-ray absorption spectroscopy with the aid of ab initio quantum-mechanical and molecular-mechanics calculations, we show how, in addition to the proton transfer, the subsequent rearrangement of the urea dimer and the associated change of the electronic structure can be identified with site selectivity. These results establish the considerable potential of flat-jet, table-top X-ray absorption spectroscopy[7,8] in elucidating solution-phase ultrafast dynamics in biomolecular systems.

Urea and its photoinduced reactions are a cornerstone in current theories on the origin of life[9]. When exposed to ionizing radiation, urea forms malonic acid[10], which further reacts with urea to yield several nucleobases[11]. The urea dimer is thus an intriguing system for understanding the first steps in the ultrafast chemistry at the origin of life and an excellent model system for studying proton transfer across hydrogen bonds, which is an important source of mutations, in both RNA and DNA[12]. Given the wide significance of urea, its dimer and photoinduced proton-transfer dynamics, this system is ideal for the application of femtosecond soft X-ray absorption spectroscopy (XAS) of liquids[8]. The recent development of table-top soft X-ray spectroscopy[3–5] has indeed opened new avenues for investigating dynamics in molecular systems with site specificity[3,13–15] and ultimate time resolution[6,16], but has so far mainly been applied to gas-phase studies. Here we investigate femtosecond proton transfer in ionized urea dimers in aqueous solution using element-specific soft X-ray transient-absorption spectroscopy. We induce the proton transfer through multiphoton ionization at 400 nm, creating large ionization fractions reminiscent of the conditions thought to prevail on the primordial earth as a consequence of the linear absorption of ionizing radiation. Whereas previous studies found that ionization-induced proton transfer between nucleobases was mediated by water[17] or can even take place in the absence of hydrogen bonds[18], we find that in ionized urea solutions proton transfer from an ionized urea donor to a urea acceptor is the dominant pathway. In addition to following the proton transfer in real time, we show through direct comparison with quantum-mechanical and molecular-mechanics (QM/MM) calculations that the underlying

dynamical evolution of the valence hole can be distinguished from the proton transfer itself. This separation of electronic and structural rearrangements is a unique feature of transient XAS. Our work thereby contributes to establish soft XAS as a promising method for explaining the broad class of non-adiabatic ultrafast dynamics in solution-phase chemistry.

Figure 1 illustrates the experimental scheme, in which a broad-band soft X-ray (SXR) probe pulse covering the carbon and nitrogen K edges (Extended Data Fig. 1) is focused on a sub-micrometre-thin liquid flat-sheet sample[19]. The transmitted SXR radiation is recorded with a spectrometer using a variable-line-spacing grating and a charge-coupled-device (CCD) X-ray camera (see Methods and ref. 8 for details). A 400 nm pulse, with a duration of approximately 30 fs serves as the pump pulse, inducing multiphoton ionization in the liquid sample at an intensity of around $1 \times 10^{14}$ W cm$^{-2}$. Lower pump intensities at both 400 nm and 266 nm were also investigated and led to the observation of the same features (Extended Data Fig. 10 and Supplementary Fig. 10) as those described here. These observations support the assignment of all observed dynamics to urea and exclude a measurable influence of plasma-induced chemistry (Supplementary Information section 1.8). Figure 1c shows the static X-ray absorption spectrum of a 10 M aqueous urea solution at the carbon and nitrogen K edges (blue) as well as the time-averaged transient absorption of the ionized sample (orange). The difference spectrum (change in optical density, ΔOD) is shown in magenta.

The carbon K-edge absorption exhibits a pre-edge feature around 290 eV, which is assigned to the C $1s \rightarrow \pi^*$ transition of urea, in agreement

[1]Laboratory of Physical Chemistry, ETH Zürich, Zurich, Switzerland. [2]GAP–Biophotonics, Université de Genève, Geneva, Switzerland. [3]Center for Free-Electron Laser Science CFEL, Deutsches Elektronen-Synchrotron DESY, Hamburg, Germany. [4]Department of Physics, Universität Hamburg, Hamburg, Germany. [5]Hamburg Centre for Ultrafast Imaging, Universität Hamburg, Hamburg, Germany. [6]Present address: International Center for Synchrotron Radiation Innovation Smart, Tohoku University, Miyagi, Sendai, Japan. [7]Present address: European XFEL, Schenefeld, Germany. [8]These authors contributed equally: Zhong Yin, Yi-Ping Chang, Tadas Balčiūnas, Yashoj Shakya. ✉e-mail: yinz@tohoku.ac.jp; ludger.inhester@desy.de; jean-pierre.wolf@unige.ch; hwoerner@ethz.ch

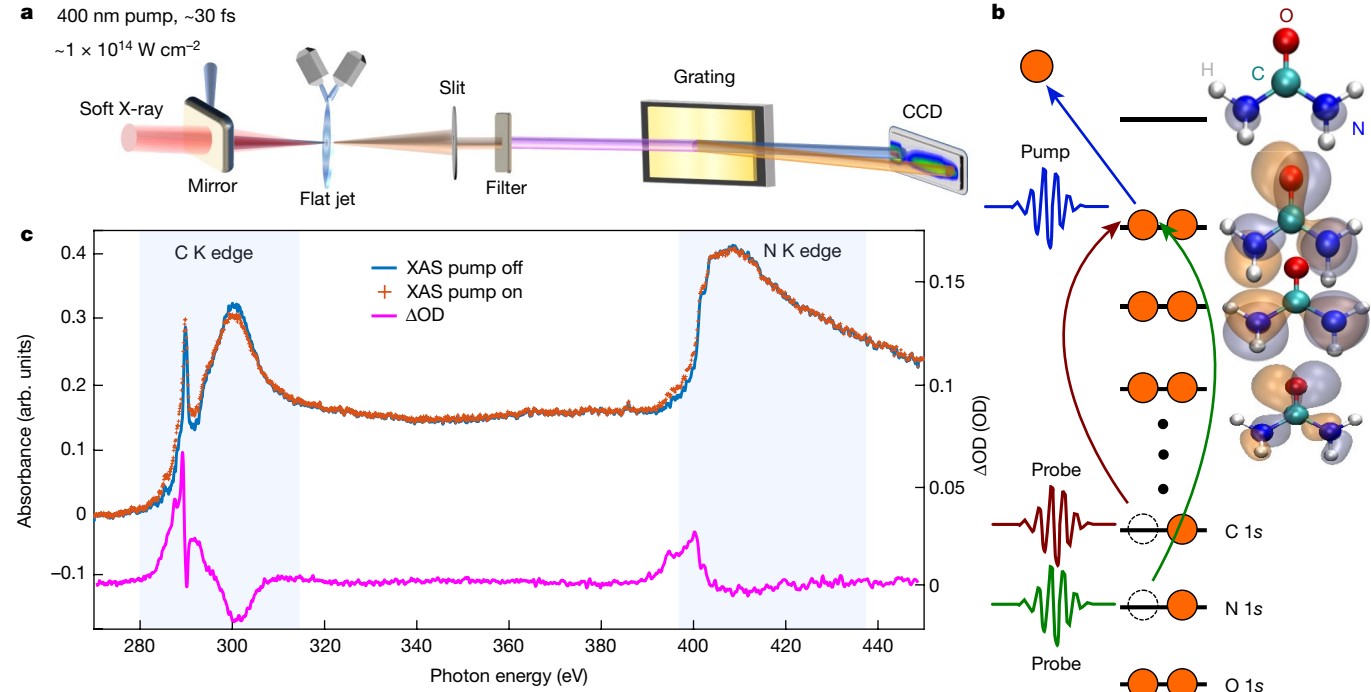

**Fig. 1 | Overview of the experimental setup and the pumped and unpumped spectra. a**, Schematic depiction of the experimental setup. **b**, Schematic molecular-orbital diagram illustrating ionization by a pump pulse, followed by probing of the system with a SXR pulse. **c**, XAS spectrum of a 10 M urea solution covering the carbon and nitrogen K edges with and without pump, and the time-averaged ΔOD signal.

with previous studies[7,20,21]. The experimental spectra have been calibrated by aligning the experimental carbon and nitrogen pre-edge peaks to previous synchrotron data[21,22]. All pre-edge absorption features of the photoionized sample originate from allowed transitions into valence vacancies created by the pump pulse.

Figure 2 shows the time-resolved ΔOD spectra of aqueous urea solutions with 10 M and 5 M concentrations recorded over a 2 ps time window. At both concentrations, the dominant features are a sharp depletion at 290 eV, corresponding to the C $1s \rightarrow \pi^*$ transition of the neutral urea molecule. The peak at 289 eV and the gradually increasing absorption feature from 285 eV to 289 eV are assigned to the C $1s \rightarrow \pi^*$ transition and C $1s$ to outer-valence vacancies of ionized urea molecules, respectively.

The main difference between the results obtained with 5 M and 10 M solutions is the appearance of an additional absorption band on top of the broad absorption feature in the latter data, which appears in the range 286.5–287 eV, shifts to around 287.5 eV and simultaneously gains in intensity (arrow in Fig. 2b). A similar feature is not observed at 5 M concentrations (Fig. 2d) nor at lower concentrations (Extended Data Fig. 9). The exclusive observation of this feature at high concentrations suggests that it originates from urea molecules linked by hydrogen bonds to other urea molecules. The abundance of urea–urea hydrogen bonds is further discussed in Supplementary Information section 1.1 based on molecular-dynamics simulations.

This feature can be identified by comparison with detailed QM/MM calculations. To that end, the valence-ionization dynamics of several urea monomer and dimer conformations were simulated using initial samples from a molecular-dynamics simulation of a 10 M aqueous urea solution and their corresponding time-resolved XAS spectra were calculated. For each trajectory, either a urea monomer together with a hydrogen-bonded water molecule or a dimer were chosen at random as the QM region. The surrounding environment was described by a force field in the ONIOM (our own *n*-layered integrated molecular orbital and molecular mechanics) scheme[23]. Non-Born–Oppenheimer effects were captured using Tully's fewest switching surface-hopping

scheme[24], with the valence-ionized system being described by Koopmans' theorem. The electronic structure calculations were performed using the XMOLECULE toolkit[25–27]. Further details on the simulations can be found in the Methods section.

After ionization, we observe that a fraction of the trajectories that use a urea dimer as QM region undergo a proton-transfer reaction (about 7% after ionization out of the highest occupied molecular orbital (HOMO) and about 17% after HOMO-3 ionization). This proton-transfer reaction resembles the one that was previously studied for a urea dimer in vacuum[28]. For the dimer trajectories that undergo proton transfer following HOMO ionization, the resulting time-resolved XAS spectra in the region below the pre-edge (that is, the region highlighted in Fig. 2a–d by dashed lines) are shown in Fig. 2e,f. The rapid oscillations in the predicted spectra originate from vibrational dynamics of the ionized urea dimers, also obtained in previous work[28]. They are not resolved in the experimental data owing to the larger delay-step size, the approximately 30 fs instrument function and the additional contribution of diffusive dynamics in the experimental data. The calculated spectra display both the increase of intensity and the energy shift of the X-ray transition observed in Fig. 2b.

The transient XAS spectra from trajectories that comprise a monomer with a hydrogen-bonded water molecule in the QM region are shown in Fig. 2g,h. In contrast to the dimer simulations, the XAS spectra from a monomer with water simulations only display a weak and nearly time-independent absorption feature, showing that ionized urea monomers do not undergo proton transfer to water. Similarly, the transient XAS spectra of ionized urea dimers that do not undergo proton transfer (shown in Supplementary Fig. 2c,d) show only minor time-dependent changes. These comparisons establish proton transfer in the ionized urea dimer as the origin of the time-dependent spectral feature highlighted with an arrow in Fig. 2b.

Figure 3a shows a magnified version of the experimental XAS spectra in the region of the proton-transfer band. These data are well represented by three Gaussian bands with time-dependent intensities shown in Fig. 3b (see also Extended Data Fig. 4). The band centred at 286.7 eV

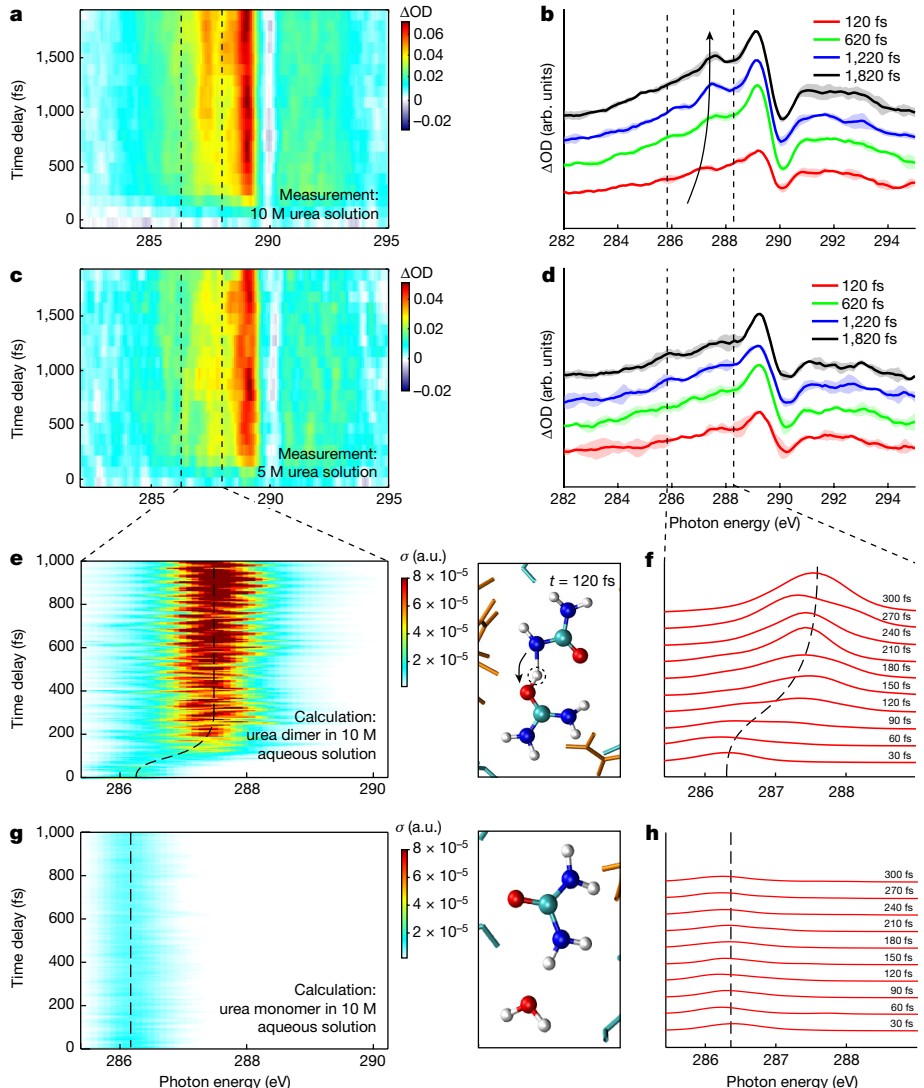

**Fig. 2 | Overview of experimental and theoretical results. a–h**, The transient carbon K-edge XAS results of 10 M or 5 M urea solutions (**a–d**) and a direct comparison with the QM/MM calculations after HOMO ionization (**e–h**) (showing cross sections in atomic units (a.u.)). Panels **a,c,e,g** show an overview of the results, and panels **b,d,f,h** show XAS spectra at selected time delays (specified in the legends). The 10 M aqueous urea solution (**a,b**) reveals the signature of proton transfer in the urea dimer, which is comparable with calculations (**e,f**). At the lower 5 M concentration (**c,d**), proton transfer is not observed in the signal fluctuations, which is supported by the lack of proton transfer in QM/MM calculations of ionized urea:water (1:1) complexes embedded in the 10 M solution (**g,h**).

decays first, followed by the band centred at 287.3 eV, after which the band centred at 287.6 eV rises.

These observations are well explained by the calculations. Whereas contributions from deeper-valence orbital ionization have previously been observed in gas-phase attosecond transient-absorption spectroscopy[6], corresponding evidence in the liquid phase has so far remained elusive. Figure 3c shows QM/MM calculations of the proton-transfer band of urea dimers following HOMO ionization. The calculations agree well with the experimental data overall, but feature a single absorption band at early times in which the experimental data show two bands that could be attributed to ionization from lower-lying orbitals. The six highest-lying molecular orbitals are located in a binding-energy range of only approximately 2 eV (Extended Data Fig. 6). Ionization from any of these six orbitals is followed by internal conversion to the electronic ground state of the solvated urea dimer cation (that is, with a hole in the HOMO) on timescales of at most 50 fs (Extended Data Fig. 7). As a representative example, we show the transient XAS spectra calculated following ionization from HOMO-3 in Fig. 3e for the trajectories that undergo proton transfer. Band i in Fig. 3f represents the

absorption in the electronically excited state of the urea dimer cation, which decays with a characteristic time of 30 ± 1 fs. Ionization from HOMO-3 thus gives rise to the appearance of a lower-lying pre-edge absorption feature, which decays more rapidly compared to the case of HOMO ionization, but leads to otherwise very similar signatures of the proton-transfer dynamics (Extended Data Fig. 8). Band i in the measured data in Fig. 3b rapidly decays in 110 ± 15 fs, which is consistent with ionization from lower-lying valence orbitals.

The experimentally measured band ii (Fig. 3b) with a decay time of 290 ± 22 fs is thus assigned to the calculated bands ii with decay times of 114 ± 2 fs and 143 ± 1 fs for the HOMO and HOMO-3 ionized dimers, respectively (Fig. 3d,f).

The 591 ± 128 fs rise time of band iii observed in the experiment is longer than in the calculations: 141 ± 2 fs and 180 ± 2 fs, following HOMO and HOMO-3 ionization, respectively. This discrepancy of timescales can be attributed to the fact that the measurements capture both real-time dynamics and delayed kinetics limited by diffusive processes, whereas the QM/MM calculations are designed to isolate the dynamics. In the simulations, proton-transfer dynamics can only take place

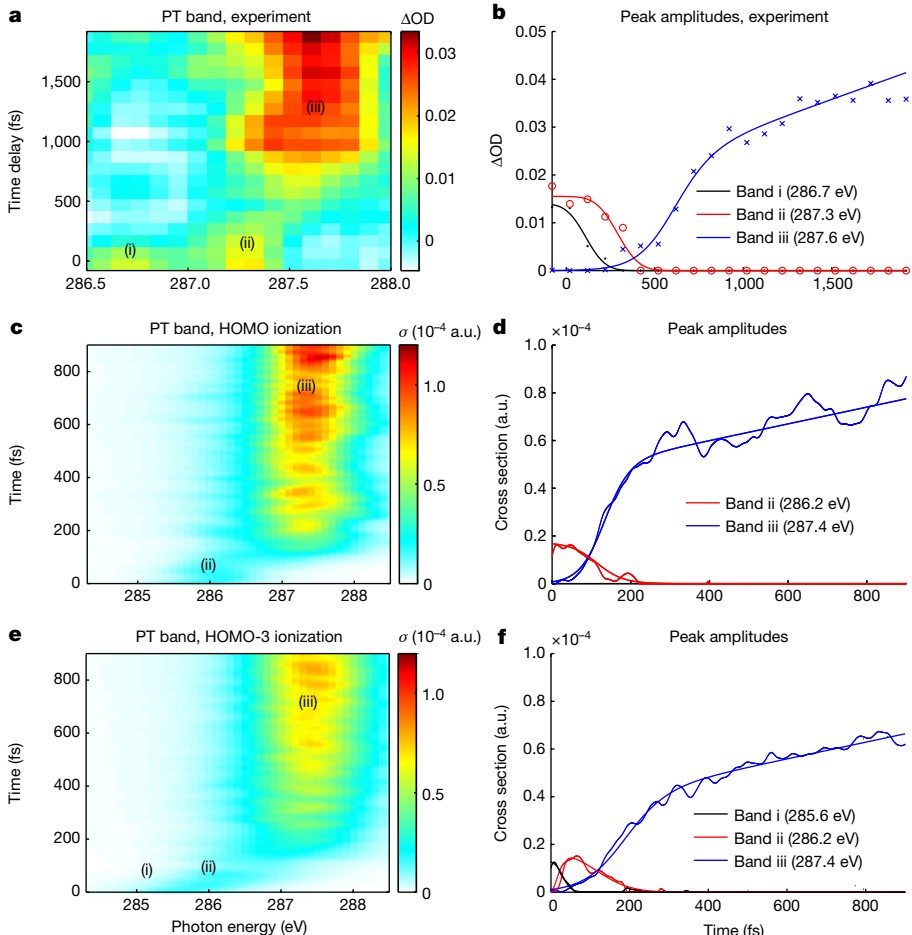

**Fig. 3 | Detailed analysis of the proton-transfer band. a,b,** Experimental data. **c–f,** QM/MM calculations. The proton-transfer (PT) bands from the experiment (**a**) and calculations for ionization of HOMO (**c**) and HOMO-3 (**e**) are analysed by fitting three Gaussians corresponding to the two pre- and one post-proton-transfer bands (i–iii) as shown in Extended Data Fig. 4. Their corresponding amplitudes are shown in **b**, **d** and **f**.

in the chosen QM region, that is, in predefined pairs of urea molecules in the relevant simulations. This fact restricts both the number and the dynamical types of proton transfer that can take place, excluding the slower proton transfers that arise from diffusion and the formation of new hydrogen bonds beyond the QM region.

We find that proton transfer is only likely to occur if the two urea molecules are linked by the hydrogen atom that lies closest to the oxygen atom ($H_{proximal}$, see Supplementary Information section 1.5 and Supplementary Fig. 4). This type of hydrogen bond occurs in the equilibrium structure of the urea dimer in vacuum[29]. By contrast, for urea dimers linked by the hydrogen bond opposite to the oxygen atom ($H_{distal}$), proton transfer is almost absent. The small fraction of urea dimers that already possess the favourable conformation will rapidly undergo proton transfer, as confirmed by the QM/MM calculations. The other urea dimers are unlikely to undergo proton transfer in the time frame of our simulations (and experiments) because diffusion is too slow and requires forming hydrogen bonds with a urea molecule outside the QM region. The time constants for the formation of $H_{proximal}$-hydrogen bonds through either local hydrogen-bond fluctuations and diffusion-controlled kinetics in MM simulations of 10 M aqueous urea solutions are $227 \pm 11$ fs and $4.1 \pm 0.1$ ps, respectively (Supplementary Information section 1.5 and Supplementary Fig. 6). This reaffirms that our QM/MM calculations capture the proton-transfer dynamics occurring on short timescales, but are not designed to describe the reaction kinetics on longer time scales induced by diffusion[30].

Proton transfer is an emblematic case of strongly coupled electronic and nuclear dynamics[1]. It is therefore interesting to explore the specific sensitivities of time-resolved XAS to such dynamics. Figure 4 presents snapshots from a QM/MM trajectory (also available as Supplementary Video 1) of the HOMO-ionized urea dimer undergoing proton transfer. After ionization, the electronic valence hole is localized on one of the urea molecules (Fig. 4a), which subsequently donates one of its protons to the neighbouring urea molecule (Fig. 4b), after which a rearrangement of the dimer geometry takes place (Fig. 4c,d). During this rearrangement, the valence hole, which initially has a vanishing density at the central carbon atom, develops a rapidly increasing amplitude near the carbon atom.

The electronic-structure rearrangement and proton-transfer processes manifest themselves in different observables in the corresponding XAS spectra. Figure 4e shows the increasing absorption cross-section of the proton-transfer band over time, which directly reflects the gradual development of electron–hole amplitude on the carbon atom. The transfer of the proton itself, in contrast, predominantly leads to a shift of the absorption band to higher photon energies by about 1.2 eV (Fig. 4f). An analysis of the time-dependent binding energies of the C 1$s$ and singly occupied molecular orbital (SOMO) of the studied trajectory reveals that the energy shift of the SOMO towards the vacuum level (that is, a decreasing binding energy) is the dominant contribution.

Interestingly, the increasing absorption cross-section (Fig. 4e) and the shifting transition energy (Fig. 4f) occur on different timescales.

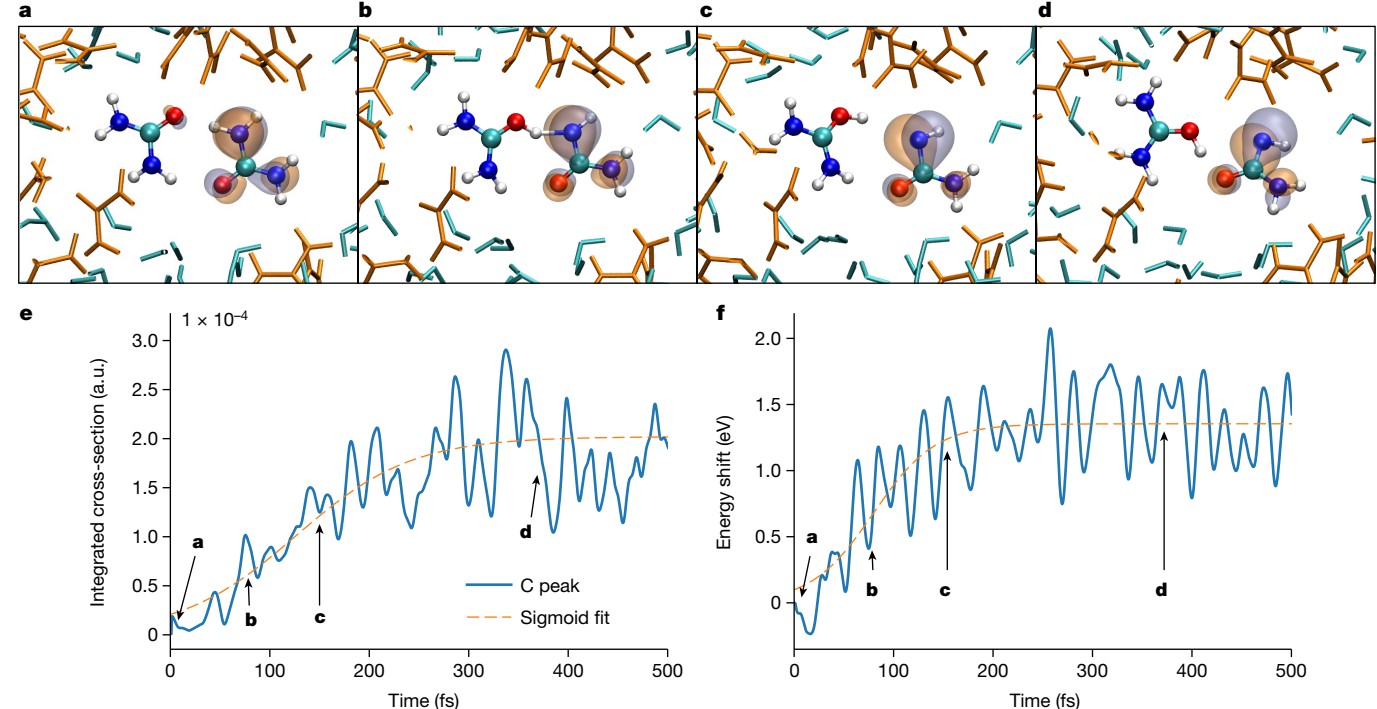

**Fig. 4 | Separation of electronic and structural rearrangements in transient XAS. a–d**, Snapshots from an exemplary QM/MM trajectory displaying proton transfer in the HOMO-ionized urea dimer along with the SOMO (or hole orbital). Time $t = 6$ fs (**a**), $t = 78$ fs (**b**), $t = 150$ fs (**c**) and $t = 372$ fs (**d**). **e**, C $1s \rightarrow$ SOMO absorption cross-section as a function of time. **f**, Energy shift of the corresponding transition. The blue lines in e and f show the calculated position and the dashed orange lines are sigmoidal fits with time constants given in the text.

A fit of the data with sigmoidal functions yields characteristic half-rise times of $126 \pm 3$ fs and $80 \pm 2$ fs for the cross section and the energy shift, respectively. Looking at the trajectory snapshots (Fig. 4a–d), these different timescales can be readily explained. The transfer of the proton is completed in Fig. 4c, corresponding to a time delay of 150 fs. At this time delay, the SOMO amplitude on the carbon atom has not yet fully developed, which is only the case after more than 300 fs (Fig. 4d). The latter timescale is longer because the electronic-structure change is driven by a rearrangement of the deprotonated urea cation to form a new hydrogen bond between the two oxygen atoms of the urea dimer, which is slower than the proton transfer itself. These conclusions derived from the analysis of a single trajectory (Fig. 4) are representative of the ensemble of calculated trajectories that undergo proton transfer (Extended Data Fig. 5). The corresponding characteristic half-rise times for the increase of the absorption strength and the shift of the transition energy amount to $134 \pm 10$ fs and $107 \pm 7$ fs, respectively.

This leads to the conclusion that time-resolved XAS in the water window is, in the present case, capable of separating the dynamical evolution of the electronic structure of solvated molecules from the effects of the proton transfer. As the sensitivity of carbon K-edge absorption strength to the time-evolving SOMO amplitude on the carbon atom has previously been observed in Rydberg-valence mixing in dissociating gas-phase $CF_4^+$ molecules[3], it is likely to be a fairly general feature. Our detailed analysis, moreover, suggests that the upward shift of the XAS transition energy can also be expected to be a generic feature of proton-transfer processes in molecular systems. These results therefore highlight the considerable potential of water-window time-resolved XAS in disentangling a dynamical evolution of the valence hole from the signature of the proton transfer itself. This leaves open the interesting question of the role of solvation in the observed dynamics. To answer this question, we show in Extended Data Figs. 5c,d simulations of an isolated urea dimer in the gas phase. In this case, we find half-rise times

of $110 \pm 3$ fs and approximately 60 fs for the absorption strength and the energy shift, respectively. Aqueous solvation thus slows down the proton transfer in the present case.

We have found that solution-phase water-window XAS applied to ionization-induced proton transfer in aqueous urea dimers selectively identifies the amplitude of the electron–hole wavefunction at the carbon atom and maps it into the absorption strength of the carbon pre-edge absorption. The transfer of the proton, in contrast, is mapped into a shift of the pre-edge absorption feature towards higher energies. These results thus establish the potential of water-window XAS for disentangling individual aspects of the respective electronic and structural dynamics in ultrafast non-adiabatic dynamics of molecular systems in a liquid environment. Ionization of prebiotically relevant urea dimers leads to an ultrafast proton transfer from one urea molecule to its neighbour, rapidly followed by a rearrangement of the electron–hole density on the donating moiety. The products of this reaction are a protonated urea molecule and a urea radical, deprived of a hydrogen atom. This reaction is remarkably similar in its nature to ionization-induced proton transfer in ionized water (bulk or clusters)[27,31,32], $H_2O \cdots HOH^+ \rightarrow H_3O^+ \cdots OH$ and in ionized alcohols[8], $ROH \cdots HOR^+ \rightarrow ROH_2^+ \cdots OR$, where R stands for the organic rest of the alcohol molecules. Our results therefore identify an interesting similarity in the earliest steps of ionization-induced ultrafast chemistry of hydrogen-bonded systems. The identified urea radical product is indeed unlikely to be stable, and will probably react further, just as OH radicals in ionized liquid water. Future work can now be envisaged to address the subsequent reaction steps and understand how malonic acid and nucleobases are being formed[10,11]. More fundamentally, solution-phase water-window XAS can now be extended to the attosecond timescale[6] to explore the possible influence of electronic coherences between electronic states created in valence ionization on the proton transfer, including a possible electronic control[33,34] of such fundamental biochemically relevant reactions.

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

# Methods

## Experimental methods

**Experimental setup.** The experimental setup consists of a laser-light source providing ultrashort laser pulses, a SXR high-harmonic source, a liquid flat-jet absorption chamber and a flat-field SXR spectrometer. The laser source consists of a titanium:sapphire laser system with an average power of 17 W, delivering approximately 40 fs pulses centred at 800 nm. This laser beam is split in a 90:10 ratio with the more intense part directed to a commercial optical parametric amplifier to generate approximately 40 fs, passively carrier-envelope-phase-stable idler pulses centred at 1,800 nm. These pulses are further compressed by means of two-stage spectral broadening in filaments and compression in bulk glass down to approximately 12 fs and used for high-harmonic generation in a high-pressure He gas target, as described in ref. 35. The weaker part of the beam is frequency-doubled in a beta-barium-borate crystal to 400 nm, focused with a 1 m focal length and recombined with the broadband SXR beam by means of a perforated mirror. For the additional wavelength-dependence measurements shown in Supplementary Fig. 10, the 266 nm pump pulse was generated by recombining the partially depleted 800 nm beam with the 400 nm pulse in a second beta-barium-borate crystal (type-I) phase-matched for sum-frequency generation. The SXR beam transmitted through the liquid jet is dispersed by a variable-line-spacing grating from Hitachi (2,400 lines mm$^{-1}$) and recorded with an X-ray CCD camera from Andor. The spectrum of the SXR probe is shown in Extended Data Fig. 1. Additional details on the experimental setup can be found in ref. 8.

The SXR spot size was estimated based on a pump drilled pinhole to be approximately 62 μm through which the probe beam intensity was higher than 90%. This is smaller than the 400 nm pump pulse with a spot size of approximately 80 μm. For details, see also the supplementary information of ref. 8.

**Sample preparation and delivery.** The various concentrated urea solutions were prepared from more than 99% purity urea samples purchased from Sigma Aldrich and dissolved in purified liquid water with an electrical resistivity of approximately 18.0 MΩ cm. The flat jet is created by collision of two cylindrical jets emanating from quartz capillaries with orifices of about 18 μm and crossed at an angle of 48°. The thickness of the flat jet in the laser/SXR propagation direction is around 500 nm. The urea solution is delivered into the interaction region by an HPLC pump. A more comprehensive description of the flat jet including sheet shape, thickness and temperature measurements can be found in refs. 19,36,37.

**Energy calibration.** X-ray absorption K edges of carbon and nitrogen from urea solutions measured at synchrotron facilites were used as reference energy points for the energy calibration in this work. The carbon K edge of aqueous urea was previously measured to be at 289.8 eV (ref. 22) and the nitrogen K edge at 402.0 eV (ref. 21).

**Temporal resolution.** Temporal resolution of our measurements was evaluated by reducing the temporal scanning step down to 10 fs. A lineout of a band 285–295 eV at the carbon edge measured using 2.5 M urea solution is shown in Extended Data Fig. 2. A fit using an error function returns a rise time of 27 fs (defined as the signal increase from 10% to 90%). Note that a much larger temporal step of 100 fs has been used for some of the measurements presented to match the timescale of the dynamics of the proton-transfer process.

**Data recording.** The SXR spectra were recorded using a Newton X-Ray CCD camera from Andor that is cooled to −95 °C during operation to reduce dark current noise. For time-resolved measurements, each time step consists of ten image acquisitions of the SXR spectra transmitted through both pumped and unpumped samples (2 × 10 total images). Each image acquisition has an exposure time of 4 s and a horizontal shift readout frequency of 50 kHz to reduce readout noise. To process the images for data analysis, we performed background subtraction, cosmic-ray filtering and spectral intensity normalization to reduce shot-to-shot spectral fluctuations.

## Theoretical computation details

**Initial molecular-dynamics setup.** The dynamics of ionized urea molecules in aqueous solution were investigated based on molecular-dynamics simulations using the hybrid QM/MM approach. To sample the possible configuration structures in the liquid phase, we first performed force-field molecular-dynamics calculations for a 10 M and a 5 M aqueous urea solution using Gromacs (v.2018.8)[38] using the GROMOS 54A7 force field[39]. The urea force field is based on ref. 40 and the SPC/E water model[41] was used. The setup is similar to the calculations in ref. 42. For the 10 M urea aqueous solution, a cubic box with volume (3 nm)$^3$ was filled with 146 urea molecules using the insert-molecules routine from Gromacs. The remaining space was then filled with 439 water molecules. For the 5 M urea aqueous solution, the same box size was used, but 81 urea molecules and 622 water molecules. The molecular-dynamics simulations were performed at a temperature of 300 K and a pressure of 1 bar using periodic boundary conditions. In particular, we used the Particle-Mesh-Ewald method (real-space cut-off parameter $r_{cutoff} = 1$ nm, Fourier spacing = 0.12 nm), Parinello–Rahman pressure coupling (time constant for pressure coupling $\tau_p = 1$ ps), velocity-rescale temperature coupling (time constant for temperature coupling $\tau = 0.1$ ps)[43] and a timestep of 0.5 fs. After 50 ps of equilibration in an ensemble with constant particle number, constant volume and constant temperature (NVT-ensemble), followed by 50 ps of equilibration in an ensemble with constant particle number, constant pressure and constant temperature (NpT-ensemble), a 1,000 ps production run (NpT-ensemble) was performed. The actual final concentrations were 10.2 M and 5.5 M. For simplicity, the two simulation setups are simply referred to as 10 M and 5 M in the text. If not stated otherwise, all the data are shown for the 10 M calculation.

From different snapshots of the 10 M simulation, we selected 149 clusters containing a hydrogen-bonded urea dimer and 150 clusters containing a urea molecule with a hydrogen-bonded water molecule. These clusters were considered as the QM region in the following simulations, whereas the rest of the molecules in the simulation box were part of MM and hence treated using classical force fields. For the selection of these clusters, we used a geometric hydrogen-bond criterion (donor–acceptor distance ≤3 Å and hydrogen donor–acceptor angle ≤20°). For each of these QM regions, the QM/MM setup was further equilibrated by a 2–ps QM/MM simulation performed with Gromacs (v.4.5.5)[44] with the same molecular-dynamics parameters as for the initial setup using the ONIOM embedding scheme[23]. In these calculations, the respective QM region was described using restricted Hartree–Fock using the 6-31+G(d)[45–47] basis set. The QM calculations were performed using the XMOLECULE package (v.3847)[25,26] with a modification of the Gromacs QM-interface code; for the evaluation of electron integrals we used the libcint library[48].

**Ionized-state simulation.** On the basis of the samples obtained from the molecular-dynamics calculations, we performed simulations with an ionized QM region. In particular, an electron was removed either from the HOMO or the deeper-valence HOMO-3 orbital. These simulations were also performed using the ONIOM embedding scheme and the same force fields for water and urea as used for the MM simulations discussed above. The QM/MM simulations were propagated in an ensemble with constant particle number, constant volume and constant energy (NVE-ensemble) setup for 1 ps with a timestep of 0.5 fs, and without imposing periodic boundary conditions (the QM region was shifted to the centre of the simulation box). The calculations were performed using a similar methodology as earlier calculations for urea and

urea dimers in vacuum[28] using Tully's fewest-switches surface-hopping scheme[24] to handle non-adiabatic dynamics. The electronic structure calculations were again performed at Hartree−Fock level with 6-31+G(d) basis set using XMOLECULE. Koopmans' theorem was used to describe the ionized QM region, which has been shown to be reasonably accurate and computationally efficient at describing valence-ionized states[27,28,49].

We have tested the accuracy of this method for the urea dimer (cyclic conformation in vacuum) by comparing the respective potential energy surfaces with those obtained with the equation-of-motion coupled cluster (EOM-CC) method using GAMESS (v.2021-R2)[50–52]. Extended Data Fig. 3 compares a cut through the potential energy surfaces for the lowest ionized states along the proton-transfer coordinate. The left panel shows potential energy surface scans for the three lowest ionized states of a urea–water complex (water is hydrogen-bonded to one of the proximal hydrogens) and the right panel shows scans for the six lowest states of the urea dimer (cyclic configuration). The geometrical configurations have been obtained through geometry optimization. For the urea–water complex, the NHO angle to the water molecule was constrained at 180°.

As can be seen, the potential energy surfaces obtained with the two methods are of similar shape. Differences occur in the relative distance of the surfaces and in the barrier heights between the two local minima. For the urea dimer (Extended Data Fig. 3, right), both methods indicate that only the ionized ground state may allow for a proton transfer, but the barrier obtained from the EOM-CC method is somewhat lower compared with the one from Koopmanns' theorem. For the urea–water complex (Extended Data Fig. 3, left), the proton transfer is prohibited for all considered ionized states by the higher lying local minimum at about 1.1 Å. For the ionized ground state, this local minimum is about 0.7 eV higher than the global minimum. Moreover, there are considerable barriers of 1.2 eV (EOM-CC) or 1.6 eV (Koopmanns' theorem). We conclude that Koopmanns' theorem is sufficient to semiquantitatively follow the ionized-state dynamics. The potential energy scans fully support the observation of the fewest-switches surface hopping simulations, showing that proton transfer from the ionized urea to water is rather unlikely. Instead, proton transfer occurs from ionized urea to a neighbouring urea if the dimer is in an appropriate configuration.

**Theoretical time-resolved X-ray absorption spectra calculation.** To compute the time-resolved X-ray absorption spectra, we calculated the absorption cross-section at each time step and for each ionized-state trajectory as described in refs. 27,28,49. In brief, for an electron in the core orbital $\varphi_i$ being excited to the valence orbital vacancy $\varphi_f$, the absorption cross-section is given by[53]

$$\sigma(\omega) = \frac{4}{3}\pi^2\omega\alpha\delta(\epsilon_f - \epsilon_i - \omega)|\langle\varphi_f|\hat{\mathbf{d}}|\varphi_i\rangle|^2 \qquad (1)$$

where $\delta(x)$ is a function describing the line shape, $\omega$ is the photon energy of the X-ray probe pulse, $\alpha$ is the fine structure constant, $\epsilon_i$ and $\epsilon_f$ are orbital energies for $\varphi_i$ and $\varphi_f$, respectively, and $\hat{\mathbf{d}}$ is the dipole vector, which in equation (1) is averaged over the three spatial dimensions. The quantities in equation (1) are given in atomic units.

To take into account effects of the MM environment in the cross-section calculations, the transition dipoles and orbital energies used in equation (1) were calculated by incorporating the MM environment via point charges defined in the force field. For the function $\delta(\epsilon_f - \epsilon_i - \omega)$ we used a finite-width line profile given by a Lorentzian function with a width of 0.5 eV to take into account the natural line width of the core-ionized state and the finite detector resolution in the experiment. The calculated absorption spectra were averaged over all ionized-state trajectories for each time step. In addition, we performed a convolution in time with a Gaussian function with a full width at half maximum of 10 fs to account for the finite time resolution in the experiment.

The absorption resonance positions here are calculated by the orbital energy difference $\epsilon_f - \epsilon_i$. Notably, the resulting energies differ by several electronvolts when compared with realistic values, which is mostly because of orbital relaxation effects following core−shell ionization not being taken into consideration in the orbital energy differences. Thus, we down-shifted the calculated absorption spectra by 15.7 eV to correct for this effect. This shift was estimated from the difference between the calculated orbital binding energies (310.21 eV) and the corrected experimental carbon K-edge ionization potentials (294.51 eV)[22]. The experimental C 1$s$ ionization potential of urea in aqueous solution was determined before by Ottosson et al.[22] as 294.0 eV. In their measurement they used the 1$b_1$ binding energy in liquid water from ref. 54, which gives a value of 11.16 eV. Recent measurements revealed, however, a somewhat higher value of 11.67 eV for this binding energy[55]. Therefore, we corrected the C 1$s$ ionization potential to 294.51 eV.

The shift of 15.7 eV is qualitatively confirmed by delta-self-consistent-field (ΔSCF) calculations, which take into account the electronic relaxation effect upon core ionization. We note, however, that the calculated resonance C 1$s$ → HOMO peak is 3.2 eV lower than the resonance position in the experimental absorption spectra. This offset can be attributed to interactions with the aqueous environment that are not covered in the current calculation setup. Therefore, all calculated spectra were shifted upwards by 3.2 eV for an easier comparison with the experimental results.

For an exemplary structural configuration taken from the molecular-dynamics simulation (urea and urea with three neighbouring water molecules), Extended Data Table 1 compares computed C 1$s$ ionization potentials. As can be seen, the ΔSCF method yields ionization potentials that are 12 eV to 13 eV lower. Furthermore, there is a clear trend that a larger basis set (6-311++G(d,p) versus 6-31+G(d)) and the incorporation of environmental water molecules yields considerably lower ionization potentials.

Having incorporated the core−hole relaxation effect, the resulting calculated absorption resonance is 3.2 eV lower than the experimental values. This offset is explained by the large effect of the aqueous environment on the valence ionization potential. By subtracting the known C 1$s$ ionization potential of aqueous urea[22] (about 294.51 eV, see correction above) from the measured C 1$s$ to valence resonance energy of 287.6 eV, we estimate a valence ionization energy of about 6.9 eV, which is about 3.4 eV lower than the ionization energy of urea in the gas phase of about 10.28 eV (ref. 56). This large effect of the aqueous environment is not captured in the calculations and is thus responsible for the too low absorption resonance energy. For an easier comparison with the experimental results, the calculated spectra in the main text have all been shifted upwards by 3.2 eV.

**Gaussian fits of proton-transfer band.** Extended Data Fig. 4 illustrates how the experimental data shown in Fig. 3a of the main text were analysed in terms of three bands (designated by i, ii and iii) to obtain the data presented in Fig. 3b. At short pump-probe delays (up to approximately 300 fs), two distinct bands (i and ii) are visible. Band i decays first, followed by band ii and finally band iii rises. These fits show that the experimental signals are well captured by the fitting model that assumes three Gaussian spectral components, each with its own time-dependent intensity.

## Data availability

All data are available at the ETH Research Collection[57].

## Code availability

All non-standard code used to analyse the data is available from the corresponding authors upon reasonable request.

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

**Acknowledgements** We thank M. Moret, A. Schneider and M. Seiler for technical support. We acknowledge funding from an ERC Consolidator Grant (Project No. 772797-ATTOLIQ), projects 200021_204844 (USCOBIX) and 200021_172946, as well as the NCCR-MUST, funding instruments of the Swiss National Science Foundation. Z.Y. acknowledges financial support from an ETH Career Seed Grant No. SEED-12 19-1/1-004952-00. Y.S. acknowledges the International Max Planck Research School for Ultrafast Imaging and Structural Dynamics (IMPRS-UFAST). L.I. and R.S. acknowledge support from the Cluster of Excellence 'CUI: Advanced Imaging of Matter' of the Deutsche Forschungsgemeinschaft (DFG)-EXC 2056-project ID 390715994.

**Author contributions** Z.Y., T.B. and Y.-P.C. carried out the experiments and analysed the experimental data. A.D., G.G. and G.F. supported the experiments. Y.S. performed the theoretical calculations under the supervision of L.I. and R.S. Y.S. and L.I. analysed the theoretical data. J.-P.W. and H.J.W. supervised the experimental work. Z.Y., T.B., Y.-P.C., Y.S., L.I., J.-P.W. and H.J.W. wrote the initial manuscript. All authors discussed the data and contributed to the manuscript.

**Funding** Open access funding provided by Swiss Federal Institute of Technology Zurich.

**Additional information**
**Correspondence and requests for materials** should be addressed to Zhong Yin, Ludger Inhester, Jean-Pierre Wolf or Hans Jakob Wörner.

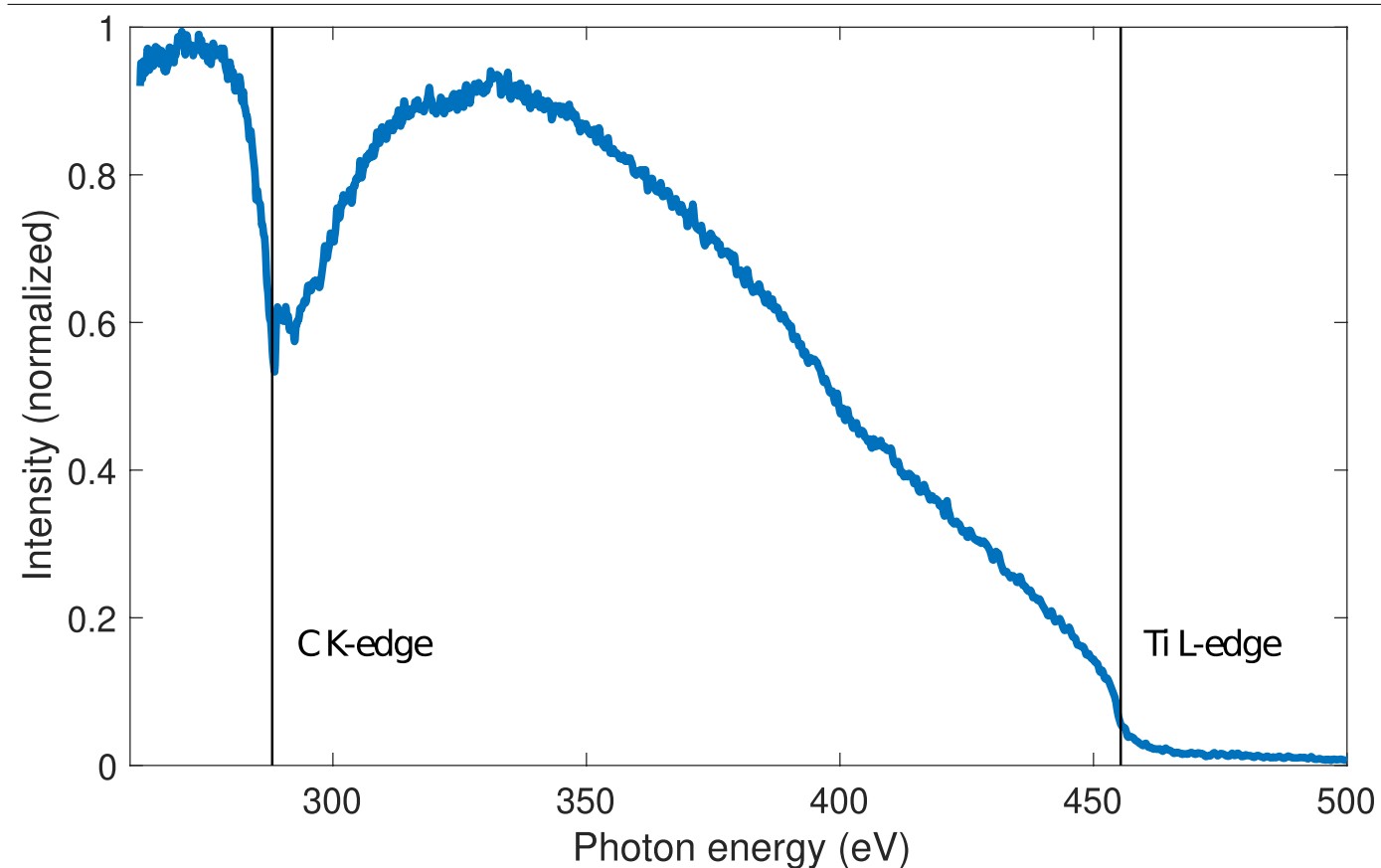

**Extended Data Fig. 1 | Broadband SXR probe spectrum.** SXR probe pulse spectrum transmitted through a 100-nm Ti filter. The spectral intensity has been corrected using the Jacobian transformation from wavelengths to photon energies. The carbon-edge absorption originates from carbon-containing contaminations on the optical components.

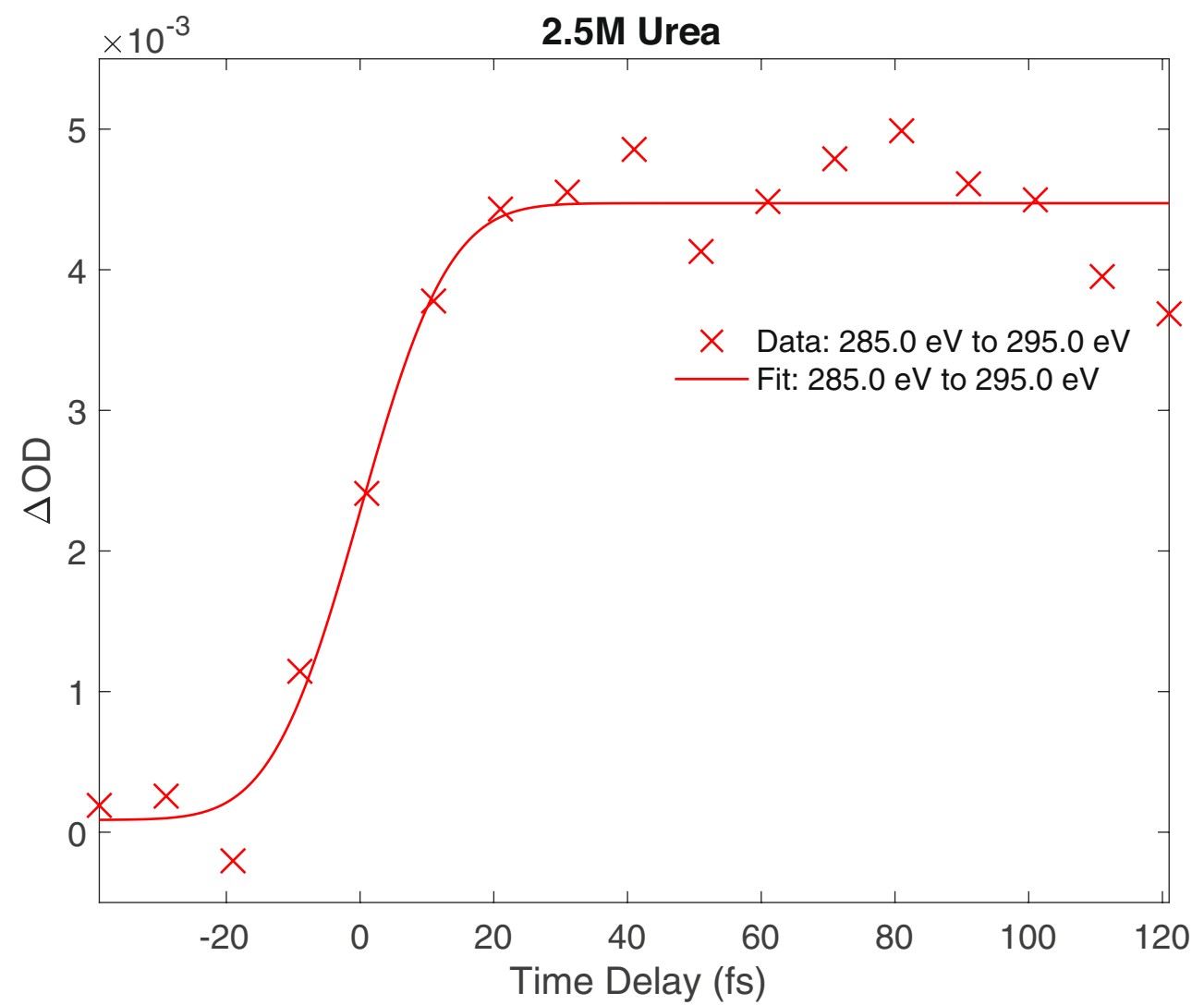

**Extended Data Fig. 2 | Rise time of the transient absorption band of the 2.5M urea solution.** Spectrally integrated differential absorbance (285–295 eV) at the carbon K-edge of a 2.5M aqueous urea solution, yielding a rise time of 27 fs (10% to 90%).

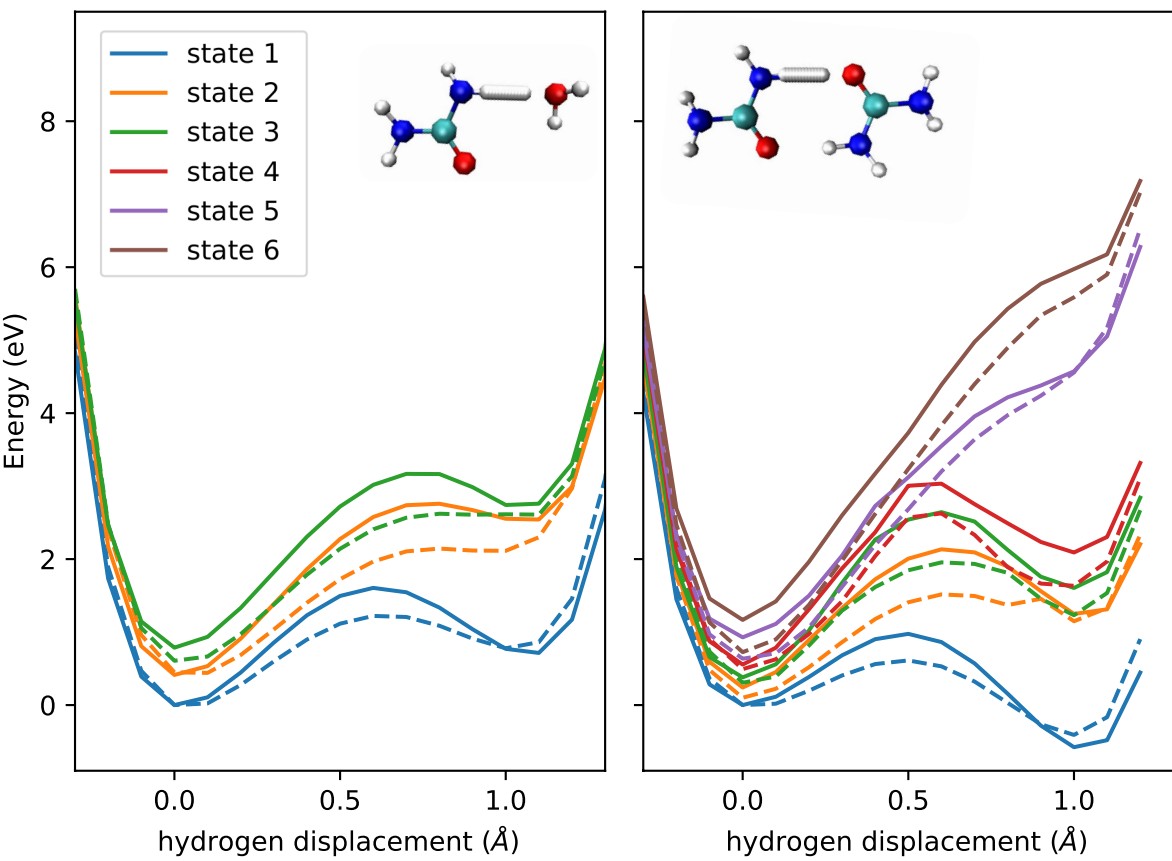

**Extended Data Fig. 3 | Potential energy surfaces from Koopmann's theorem in comparision with the EOM-CC method.** Scan through the potential-energy surfaces of the lowest ionized states along the proton transfer coordinate for a urea water complex and a urea dimer (cyclic conformation) in vacuum. The solid lines show results obtained using Koopmanns' theorem, the dashed lines show results obtained using the EOM-CC method. The x-axis indicates displacement of the proton from the neutral ground state equilibrium geometry towards the acceptor oxygen.

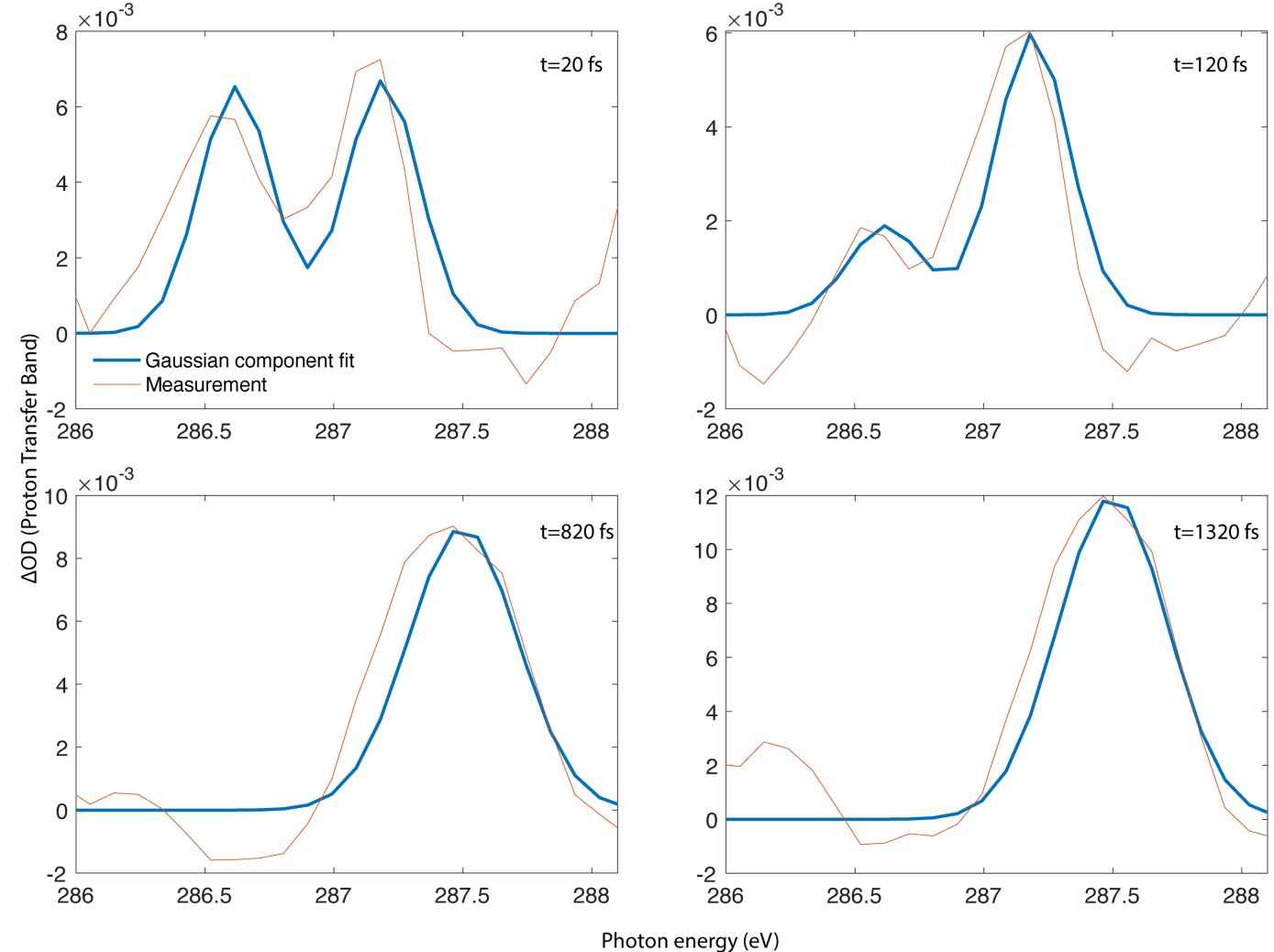

**Extended Data Fig. 4 | Fitting of the proton transfer band.** Fits of Gaussian profiles of the proton transfer band at four different pump-probe time delays.

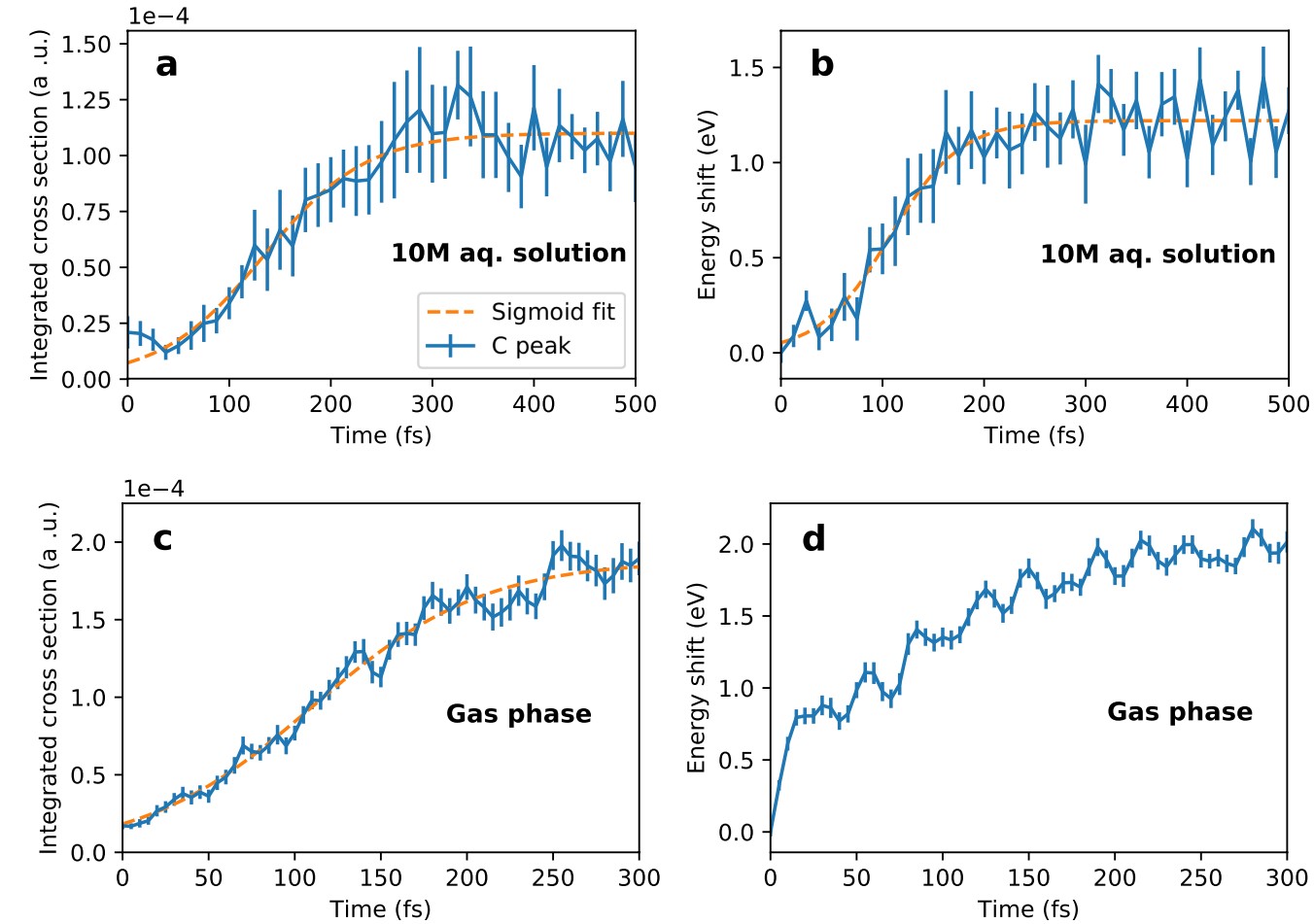

**Extended Data Fig. 5 | Proton transfer dynamics of ionised cyclic urea dimer in solution in comparison with gas phase.** (a) Ensemble-averaged C 1s→valence resonance integrated absorption cross section and (b) energy shift of the corresponding transition as a function of time along with error bars for the QM dimer trajectories in 10 M solution that undergo proton transfer following HOMO ionization. For comparison, subfigures (c) and (d) show the same quantities for a cyclic dimer in vacuum from ref. 28 (please note the different time scale here). The blue line shows the calculated values and the dashed orange line is a sigmoidal fit with time constants given in the text.

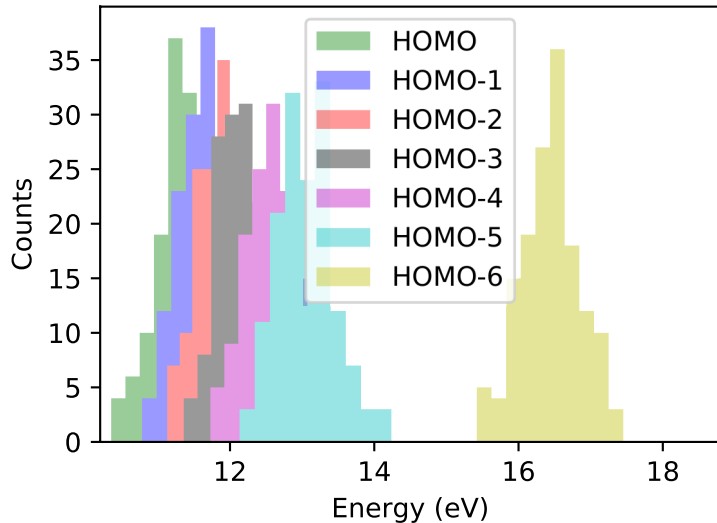

**Extended Data Fig. 6 | Theoretical binding energies of urea dimers.** Histogram of the calculated molecular-orbital binding energies from HOMO to HOMO-6 for the initial QM/MM ensemble of the neutral QM urea dimer in 10 M aqueous solution.

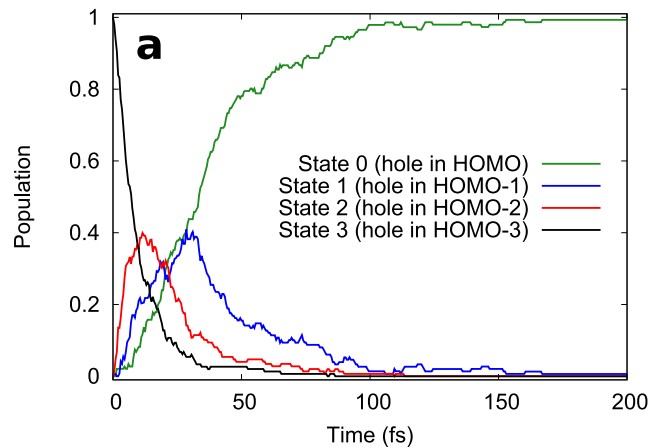
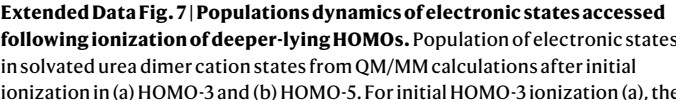
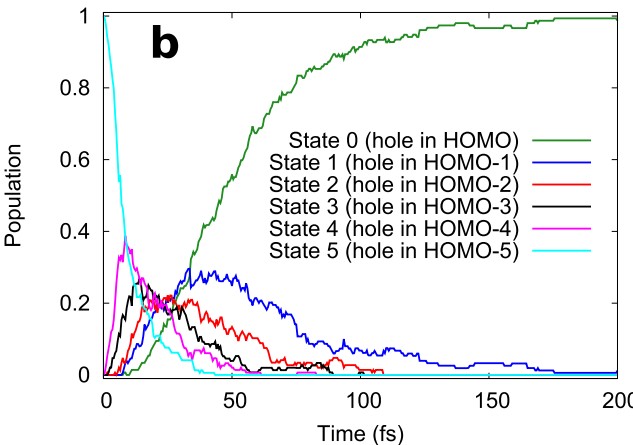

**Extended Data Fig. 7 | Populations dynamics of electronic states accessed following ionization of deeper-lying HOMOs.** Population of electronic states in solvated urea dimer cation states from QM/MM calculations after initial ionization in (a) HOMO-3 and (b) HOMO-5. For initial HOMO-3 ionization (a), the half decay time obtained by an exponential fit to state 3 (hole in HOMO-3) is $(6.68 \pm 0.02)$ fs and the half rise time obtained by a sigmoidal fit to state 0 population (hole in HOMO) is $(34.19 \pm 0.10)$ fs.

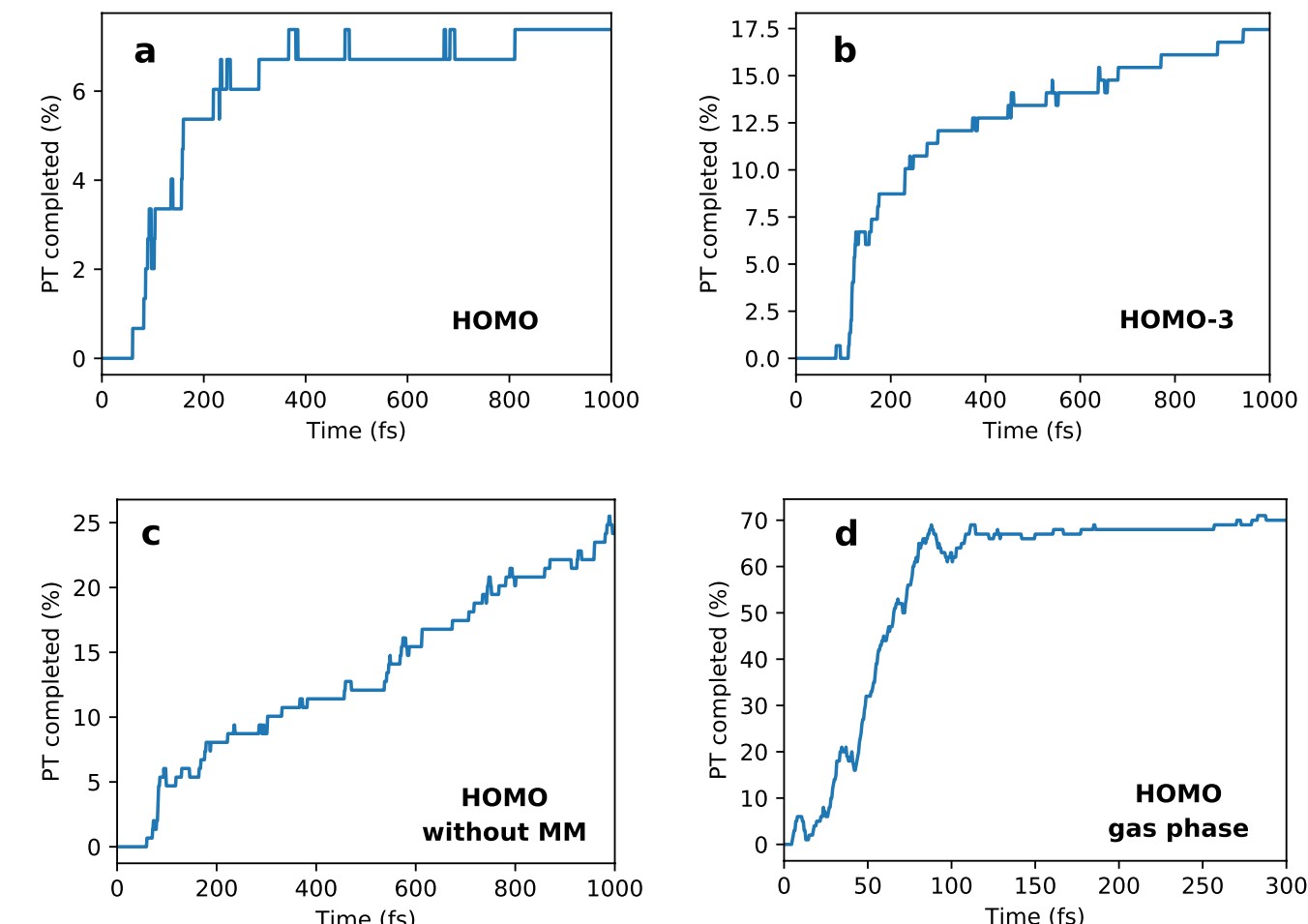

**Extended Data Fig. 8 | Proton transfer fractions of urea dimers after ionization.** Percentage of completed proton transfer within urea dimer following ionization of (a) HOMO, (b) HOMO-3 and (c) HOMO without MM environment in a 10 M aqueous urea solution, and (d) HOMO ionization of gas-phase cyclic dimer in vacuum from ref. 28 (please note the different time scale here).

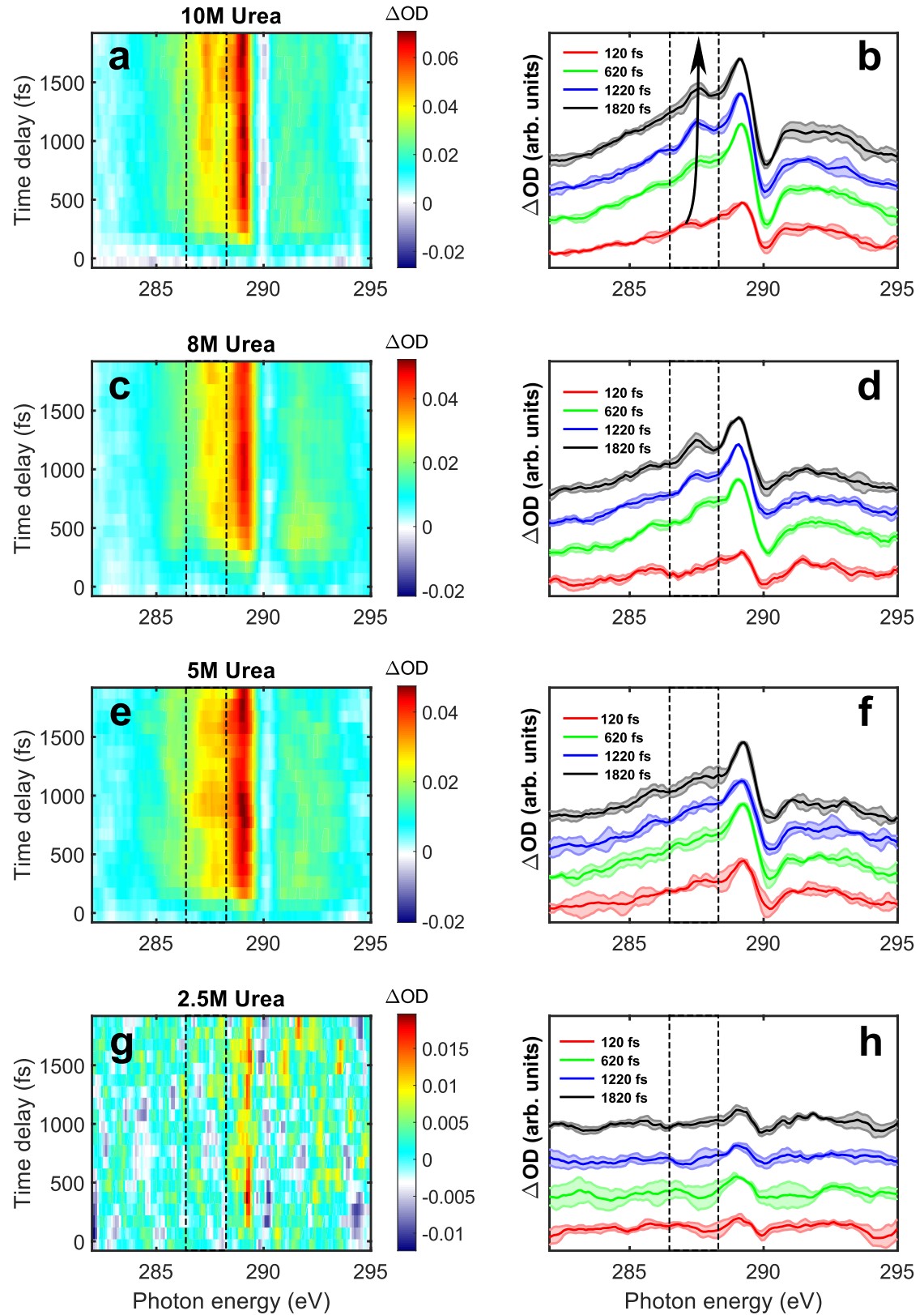

**Extended Data Fig. 9 | Concentration dependence measurement of urea solutions.** Time-resolved XAS at the carbon K-edge of 10M, 8M, 5M and 2.5M urea aqueous solutions, recorded under conditions otherwise identical to Fig. 2 of the main text.

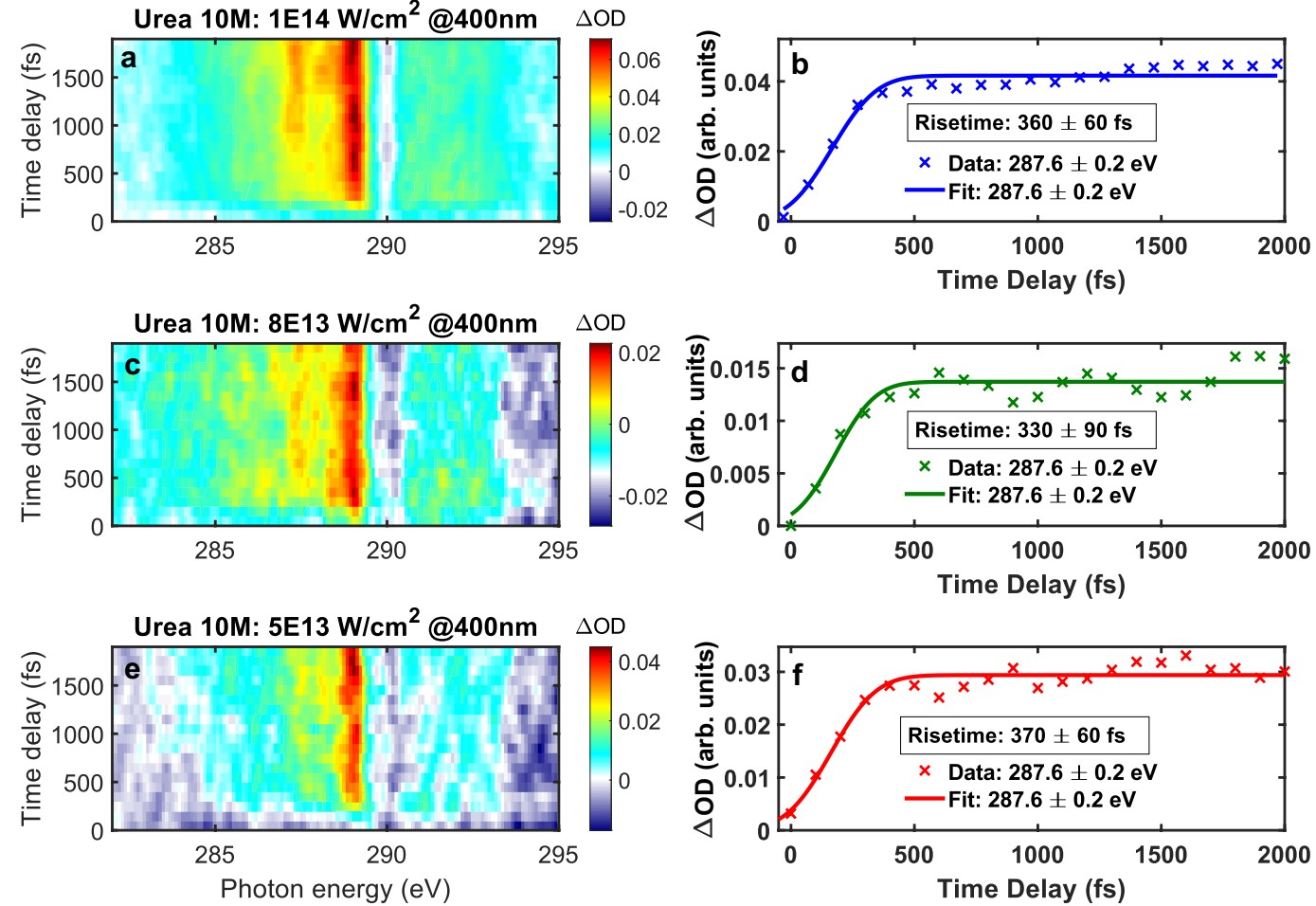

**Extended Data Fig. 10 | Pump-intensity dependence.** Time-resolved XAS at the carbon K-edge of 10M aqueous urea solutions performed with ~30 fs pump pulses centered at 400 nm and variable peak intensities. The rise times of the raw (no background subtraction) proton transfer bands obtained from each measurement are given in panel (b),(d) and (f) respectively.

**Extended Data Table 1 | Calculated C 1s ionization potentials for an exemplary conformation from the MD simulations**

| Molecule | Basis Set | C1s Ionization Potential from | | |
|---|---|---|---|---|
| | | Orbital Energy (eV) | ΔSCF (eV) | Difference (eV) |
| urea | 6-31+G(d) | 309.90 | 298.15 | 11.75 |
| | 6-311++G(d,p) | 310.04 | 296.63 | 13.31 |
| urea + 3 water | 6-31+G(d) | 310.20 | 298.27 | 11.94 |
| | 6-311++G(d,p) | 310.06 | 296.60 | 13.47 |

The QM region is either urea or urea with 3 neighboring water molecules. In the calculation, the remaining environment was embedded via point charges.