## [Peer Review File · Nature]

Manuscript Title: Femtosecond Proton Transfer in Urea Solutions Probed by X-ray Spectroscopy

Reviewer Comments & Author Rebuttals

Reviewer Reports on the Initial Version:

Referee #1 (Remarks to the Author):

This is a nice work illustrating the power of X-ray spectroscopy. The system (aqueous urea) is simple yet interesting because its role in the prebiotic chemistry on Earth, as nicely explained in the introduction.

The setup is as follows: the aqueous solution of urea is ionized to produce an unspecified valence-hole state, and then probed by time-resolved XAS on the carbon and nitrogen edge. The TR-XAS spectra are, as always, not well resolved. The authors focus on the increase in the intensity and spectral shift of the main transition. They claim that the former represents electronic dynamics and the latter -- structural change. The interpretation of the results is based on extensive calculations, which makes the story convincing. Overall, I think this is a very interesting work, worthy of publication in Nature, however, several issues should be addressed.

First, I am not sure I am convinced by the demarcation between the electronic and structural dynamics that the authors claim they can distinguish. Sure, proton transfer is a structural process, however, can one really claim that the orbital relaxation is completely uncoupled from the short-time nuclear dynamics (molecular vibrations and changes in bond-lengths induced by ionization)? This statement makes a nice story, but I think it is a bit of an oversimplification.

Second, I am struggling with the following question--why there no PT in urea-water dimers? It is indeed well known that ionization "acidifies" matter. The ionization-induced PT has been documented by many studies, for example, in synchrotron photoionization efficiency measurements (such as those carried out at LBNL) there are always copious amounts of protonated species.

Ionization-induced proton transfer in dimers of nucleobases have been studied and reported in these papers: J. Phys. Chem. A 117, 6789 (2013) and Nature Chemistry 4, 323 (2012), which I think should be mentioned and the findings reported here should be contrasted to the previous findings.

In the JPCA paper, PT between nucleobases was shown to be mediated by water molecules. In the Nature Chem. paper, it was shown that efficient PT can occur *without* hydrogen bonds, i.e., between methylated uracils. These studies lacked time resolution, but were supported by high-level calculations.

I would like to see a more detailed analysis of this issue--why there is no PT from urea to water. The energy diagram in the supplemental (Fig. S3) is very useful. Perhaps, the issue of PT to water can be clarified by constructing a similar diagram for the PT from urea to water?

I am fully on board with the authors praising the power of table-top X-ray instruments. However, they should acknowledge parallel developments by other researches. For example, Steve Leone group has published quite a few papers on their carbon edge table top studies, including time resolved ones, such as this one: Nature Comm. 12, 5003 (2021). I understand that the length of the paper is constrained by the journal requirements, however, an effort should be made to acknowledge the work of others.

Miscellaneous:

1. What is Wobble base pairs? I never heard of this term.
2. The authors use the word 'resonance' (e.g., on page 3: 'energy shift of XAS resonance') to refer to the energy of XAS transition, which is confusing. I suggest to use accepted terminology. In the same vein, page 13: "upward shift of the XAS resonance" would commonly be referred to as "blue shift of the XAS transition energy".
3. Page 4: the authors say that the main XAS transitions in urea are of $C(1s) \rightarrow \pi^*$ and $C(1s) \rightarrow \sigma^*$ types, quoting early studies. I strongly suspect this is an erroneous assignment, based on low-level calculations and mistakes in the interpretation. The π^* transition is fine, by the second one is likely to be of the Rydberg type (as σ^* would appear much higher and with lower oscillator strength). See for example this analysis of the XAS in benzene, which corrects earlier mistakes of this type: J. Phys. Chem. A 124, 9532 (2020). The nature of the transition can be easily checked by calculations using basis set with diffuse functions and NTO analysis (make sure to adjust the isovalue to much smaller values to correctly visualize Rydberg states).
4. The authors use 'at.u' to denote atomic units. The standard notation is 'a.u.'
5. Fig. 2 caption: The authors refer to concentration being 10 M in the calculations. Is this really so? If yes, they should explain in the SI how concentration in simulations was controlled.
6. Page 6: "a peak at 289 eV, assigned to the corresponding transition of the urea cation, and a broader, gradually increasing absorption feature extending from 285 eV to 289 eV. This region is assigned to vacancies created by outer-valence ionization of urea molecules." I am confused by this sentence -- is the region 285-289 also due to cation, as the last sentence suggests? If so, this part should be reworded for clarity.
7. Page 10: I am not sure I am convinced by the explanation of the mismatch in time scales for PT from the theory and experiment. If diffusion is involved, then the experiment should show multiple time-scales -- the fast one for those structures that are aligned for PT and slow one that are diffusion controlled? Could it be that the mismatch is due to relatively low level of theory used to model this process?
8. Page 11: "Proton transfer is an emblematic case of strongly coupled electronic and nuclear dynamics." A citation to Hammes-Schiefer work would be appropriate here.

9. In SI, page 9, when discussing deficiencies of Koopmans treatment of XAS: the main reason why orbital energy differences are so much off is because Koopmans theorem neglects Coulomb interaction between the hole and the particle. These energies are usually on the order of 10+ eV. This can be easily validated either by direct calculation of the corresponding integral, or by comparing CIS calculation versus Koopmans estimate.

Referee #2 (Remarks to the Author):

The manuscript by Z. Yin et al. reports the observation of photo-induced molecular dynamics in biology-related molecular dimers in solvent by laser-based soft-X-ray (SX) transient absorption spectroscopy (TAS). Experimentally, they used a table-top femtosecond SX source to measure transient absorption at carbon K edge up to a few picoseconds. The solvent target is prepared in the form of a flat liquid jet, which is very novel and original to the authors' group. Extensive quantum mechanical and molecular mechanics simulations allow to assign three bands at carbon K edge, and the observed shift of the peak position and absorption change are successfully attributed to proton transfer and molecular conformation respectively, with slightly different time scales.

Their discussions are highly convincing, illuminating the general features of intermolecular proton transfer and associated molecular structural changes. These observations cannot be done by conventional ultrafast spectroscopy with the visible light, and clearly showing the versatility of laser-based SX-TAS. This report will become a milestone in attosecond chemistry, which is emerging in the past several years after the development of attosecond technologies in the past two decades. I consider that this paper will meet the general scientific interests of the journal Nature.

I thus recommend this manuscript to be published with minor revisions, provided that the authors answer to the following technical questions.

(1) In Fig. 3, the bands (i) and (ii) shows rapid decays, but the signals are revived after 1500 fs. These revivals are absent in the panel B. The authors should correct the panel B or explain this feature.

(2) The authors claim that the oscillative features of Fig.2(E) is absent in the experimental data in Fig.2(A) because the scanning step is larger than the oscillation period. However, the sampling time is much shorter, and thus, the oscillation will be reflected in the experimental data as a fluctuation of the peak positions. This point should be clarified.

(3) It is not clear how the experimental data shown in Fig. 3(B) corresponds to Fig. S9. At first thought, these two panels have a good correspondence, but the time scales are very different. Please clarify.

(4) Regarding the experimental conditions, there is no explanation on the spot sizes for SX probe and visible pump beams. Since the pump process is highly nonlinear, I suppose that the pump spot is smaller than that of the probe SX. This point should be clarified.

Referee #1 (Remarks to the Author):

This is a nice work illustrating the power of X-ray spectroscopy. The system (aqueous urea) is simple yet interesting because its role in the prebiotic chemistry on Earth, as nicely explained in the introduction.

We thank the referee for these supportive comments.

The setup is as follows: the aqueous solution of urea is ionized to produce an unspecified valence-hole state, and then probed by time-resolved XAS on the carbon and nitrogen edge. The TR-XAS spectra are, as always, not well resolved. The authors focus on the increase in the intensity and spectral shift of the main transition. They claim that the former represents electronic dynamics and the latter -- structural change. The interpretation of the results is based on extensive calculations, which makes the story convincing. Overall, I think this is a very interesting work, worthy of publication in Nature, however, several issues should be addressed.

First, I am not sure I am convinced by the demarcation between the electronic and structural dynamics that the authors claim they can distinguish. Sure, proton transfer is a structural process, however, can one really claim that the orbital relaxation is completely uncoupled from the short-time nuclear dynamics (molecular vibrations and changes in bond-lengths induced by ionization)? This statement makes a nice story, but I think it is a bit of an oversimplification.

We thank the referee for this excellent question. We agree with the referee that nuclear displacements drive electronic-structure changes, such that the two types of dynamics are in general coupled. We did not intend to negate the existence of such couplings in the present case. What might have led to confusion with our initial formulation is that we did not clearly explain what we called "electronic dynamics" and what drives it in the present case.

What we called "electronic dynamics" is the time-dependent change in the shape of the electron hole. This is what causes the exceptionally large increase of the absorption at the carbon K-edge (see Fig. 4 in the main text). This dynamical evolution of the electronic structure (hole shape) is driven by a rearrangement of the deprotonated urea cation to form a new hydrogen bond between the two oxygen atoms of the urea dimer. The change in the shape of the electron hole is thus the consequence of a structural rearrangement. However, similar changes of the electronic structure can also be driven by purely electronic dynamics of coupled electronic-nuclear (electronically non-adiabatic) dynamics.

In contrast to the absorption strength, the large energy shift of the studied X-ray transition originates from a change in the chemical shift of the C1s binding energy. This shift occurs in direct response to the proton moving away from the carbon atom. The energy position of the X-ray transition is thus directly sensitive to the proton transfer (i.e. structural dynamics) and not to the shape of the electron hole.

We have tried to clarify these aspects by making the following changes to the manuscript:
in the abstract:

previous version:

we show how electronic-structure changes can be identified with atomic resolution and separated from the effects of the proton transfer.

new version:

we show how in addition to the proton transfer, the subsequent rearrangement of the urea dimer and the associated change of the electronic structure can be followed with site selectivity.

on p.3

previous version:

In addition to following the proton transfer in real time, we show through direct comparison with quantum mechanics/molecular mechanics (QM/MM) calculations that the underlying **electronic and nuclear dynamics** can be distinguished.

new version:

In addition to following the proton transfer in real time, we show through direct comparison with quantum mechanics/molecular mechanics (QM/MM) calculations that the **subsequent chemical dynamics causing a dynamical evolution of the valence hole** can be distinguished from the proton transfer itself.

on p. 11-12

previous version:

At this time delay, the SOMO amplitude on the carbon atom has not yet fully developed, which is only the case after more than 300~fs (see panel d).

new version:

At this time delay, the SOMO amplitude on the carbon atom has not yet fully developed, which is only the case after more than 300~fs (see panel d). **This longer time scale is caused by the fact that the electronic-structure change is driven by a rearrangement of the deprotonated urea cation to form a new hydrogen bond between the two oxygen atoms of the urea dimer, which is slower than the proton transfer itself.**

p.13

previous version:

These results therefore highlight the considerable potential of water-window time-resolved XAS in disentangling **electronic and nuclear dynamics** of solvated molecules.

new version:

These results therefore highlight the considerable potential of water-window time-resolved XAS in disentangling **a dynamical evolution of the valence hole driven by nuclear dynamics, from the signature of the proton transfer itself.**

in the conclusion:

previous version:

We have applied solution-phase water-window XAS to resolve **the electronic and structural** dynamics accompanying femtosecond proton transfer in aqueous urea dimers. (...)

These results thus establish the potential of water-window XAS for disentangling **electronic and structural dynamics** in ultrafast non-adiabatic dynamics of complex systems in a liquid environment.

new version:

We have applied solution-phase water-window XAS to resolve **different types of** dynamics accompanying femtosecond proton transfer in aqueous urea dimers. (...)

These results thus establish the potential of water-window XAS for disentangling individual aspects of the respective electronic and structural dynamics in ultrafast non-adiabatic dynamics of complex systems in a liquid environment.

Second, I am struggling with the following question--why there no PT in urea-water dimers? It is indeed well known that ionization "acidifies" matter. The ionization-induced PT has been documented by many studies, for example, in synchrotron photoionization efficiency measurements (such as those carried out at LBNL) there are always copious amounts of protonated species.

Ionization-induced proton transfer in dimers of nucleobases have been studied and reported in these papers: J. Phys. Chem. A 117, 6789 (2013) and Nature Chemistry 4, 323 (2012), which I think should be mentioned and the findings reported here should be contrasted to the previous findings. In the JPCA paper, PT between nucleobases was shown to be mediated by water molecules. In the Nature Chem. paper, it was shown that efficient PT can occur *without* hydrogen bonds, i.e., between methylated uracils. These studies lacked time resolution, but were supported by high-level calculations.

I would like to see a more detailed analysis of this issue--why there is no PT from urea to water. The energy diagram in the supplemental (Fig. S3) is very useful. Perhaps, the issue of PT to water can be clarified by constructing a similar diagram for the PT from urea to water?

We agree with the referee that this is an interesting and important question.

We have extended Fig. S3 as the reviewer suggested. The revised figure now also shows a scan (Koopmans' theorem and EOMCC) of the potential energies of the lowest ionized states along the proton-transfer coordinate from urea to water. The two PES scans now clearly demonstrate that proton transfer to another urea is much more favorable than proton transfer to water. The text discussing Fig. S3 in the SI has been changed accordingly.

We additionally point out that the proton affinity of urea (868.4 kJ/mol) is considerably higher than the one of water (691 kJ/mol). The same also holds for the gas-phase basicity (839 kJ/mol vs. 660 kJ/mol). These urea values are reported in J. Phys. Chem. A 106, 9939; Water values are from the NIST database. A proton transfer from urea to water is therefore less likely to happen.

We have also added the two references mentioned above (new Refs 17, 18) and the following text (p. 3):

"Whereas previous studies found that ionization-induced proton transfer between nucleobases was mediated by water [17] or can even take place in the absence of hydrogen bonds [18], we find in ionized urea solutions proton transfer from an ionized urea donor to a urea acceptor is the dominant pathway."

I am fully on board with the authors praising the power of table-top X-ray instruments. However, they should acknowledge parallel developments by other researches. For example, Steve Leone group has published quite a few papers on their carbon edge table top studies, including time resolved ones, such as this one: Nature Comm. 12, 5003 (2021). I understand that the length of the paper is constrained by the journal requirements, however, an effort should be made to acknowledge the work of others.

Our initially submitted manuscript already acknowledged the work of other research groups on table-top X-ray instruments, in particular the work of Steve Leone et al (formerly refs. 9 and 11, now

refs. 4 and 13), Jiro Itatani et al (formerly ref. 10, now ref. 5) and Arnaud Rouzée et al. (formerly ref. 14, now ref. 7).

We have now additionally added the above reference as new Ref. 15 in the revised manuscript.

Miscellaneous:

1. What is Wobble base pairs? I never heard of this term.

The term “Wobble base pair” describes the base pairing that occurs between nucleotides of RNA molecules and does not follow the Watson-Crick base pairing rule of DNA. Wobble base pairs are guanine-uracil (G-U), hypoxanthine-adenine (I-A), hypoxanthine-uracil (I-U) and hypoxanthine-cytosine (I-C).

We have clarified this point by rewriting the corresponding sentence as follows (p. 2), keeping the editorially requested brevity in mind:

“equivalent to those found in **the Wobble base pairs, which are responsible for the secondary structure of ribonucleic acid (RNA)** and the proper translation of the genetic code.”

2. The authors use the word 'resonance' (e.g., on page 3: 'energy shift of XAS resonance') to refer to the energy of XAS transition, which is confusing. I suggest to use accepted terminology. In the same vein, page 13: "upward shift of the XAS resonance" would commonly be referred to as "blue shift of the XAS transition energy".

We have changed “absorption resonance” to “X-ray transition energy” throughout the manuscript.

3. Page 4: the authors say that the main XAS transitions in urea are of C(1s)->pi* and C(1s)->sigma* types, quoting early studies. I strongly suspect this is an erroneous assignment, based on low-level calculations and mistakes in the interpretation. The pi* transition is fine, by the second one is likely to be of the Rydberg type (as sigma* would appear much higher and with lower oscillator strength). See for example this analysis of the XAS in benzene, which corrects earlier mistakes of this type: J. Phys. Chem. A 124, 9532 (2020). The nature of the transition can be easily checked by calculations using basis set with diffuse functions and NTO analysis (make sure to adjust the isovalue to much smaller values to correctly visualize Rydberg states).

We agree with the referee that the C(1s) → sigma* assignment is uncertain, and possibly unlikely. In our case, it was taken from previous literature.

We have therefore removed this particular assignment.

We have added the recommended citation as the new ref. 14.

4. The authors use 'at.u.' to denote atomic units. The standard notation is 'a.u.'

We have changed ‘at.u.’ to ‘a.u.’ throughout the manuscript and SM.

5. Fig. 2 caption: The authors refer to concentration being 10 M in the calculations. Is this really so? If yes, they should explain in the SI how concentration in simulations was controlled.

How the concentrations were realized in the calculations is explained in the SM, Section 1.2.1, p. 5:

“For the 10 M urea aqueous solution, a cubic box with volume $(3 \text{ nm})^3$ was filled with 146 urea molecules using the insert-molecules routine from Gromacs. The remaining space was then filled with 439 water molecules. For the 5 M urea aqueous solution, the same box size was used, but 81 urea molecules and 622 water molecules. The MD simulations were performed at a temperature of 300 K and a pressure of 1 bar using periodic boundary conditions.”

During the ionized state calculations, we employ a layer of MM molecules so that the local concentration in the QM region and its vicinity does not change and therefore remains the same as in the initial sampling.

6. Page 6: "a peak at 289 eV, assigned to the corresponding transition of the urea cation, and a broader, gradually increasing absorption feature extending from 285 eV to 289 eV. This region is assigned to vacancies created by outer-valence ionization of urea molecules." I am confused by this sentence -- is the region 285-289 also due to cation, as the last sentence suggests? If so, this part should be reworded for clarity.

We thank the referee for requesting this clarification.

We have rewritten this section to read (p. 6):

“The peak at 289 eV and the broader, gradually increasing absorption feature extending from 285 eV to 289 eV are assigned to the $C1s \rightarrow \pi^*$ transition and $C1s$ to outer-valence-vacancies of ionized urea molecules, respectively.”

7. Page 10: I am not sure I am convinced by the explanation of the mismatch in time scales for PT from the theory and experiment. If diffusion is involved, then the experiment should show multiple time-scales -- the fast one for those structures that are aligned for PT and slow one that are diffusion controlled? Could it be that the mismatch is due to relatively low level of theory used to model this process?

This point is also very well taken. Figure S14 does indeed show the two time scales that are correctly described by the referee. The experimental data shown in Fig. 3B is also consistent with this prediction because it shows a fast rise up to 700-800 fs, followed by a slower rise on the picosecond time scale. Since the experimentally observed dynamics are clearly not well represented by simple (double-exponential) kinetic laws, we have refrained from trying to extract two time constants from the experimental data.

8. Page 11: "Proton transfer is an emblematic case of strongly coupled electronic and nuclear dynamics." A citation to Hammes-Schiefer work would be appropriate here.

We thank the referee for reminding us of the seminal work of Sharon Hammes-Schiefer on proton transfer.

We have added one of her representative publications as new Ref. 1 to the revised manuscript: <https://aip.scitation.org/doi/10.1063/1.467455>

9. In SI, page 9, when discussing deficiencies of Koopmans treatment of XAS: the main reason why orbital energy differences are so much off is because Koopmans theorem neglects Coulomb interaction between the hole and the particle. These energies are usually on the order of 10+ eV. This can be easily validated either by direct calculation of the corresponding integral, or by comparing CIS

calculation versus Koopmans estimate.

The referee is correct in pointing out that estimating excitation energies from orbital energy difference between hole and particle orbital energies is lacking contribution due to the particle-hole interaction. As they remark, this particle-hole interaction term is incorporated by, e.g., the configuration interaction singles (CIS) method. However, the referee has overlooked here that we are addressing ionized states.

Koopmans' theorem applies to these ionized states (but not to excited states for which CIS would be more appropriate, indeed). It incorporates particle-hole interaction terms through the mean field of the common neutral reference. This is, of course, limited to the situation that the x-ray excitation re-occupies a previously ionized valence vacancy. We further refer to our discussion in SI 1.2.3. By comparing to the core-ionization potential obtained from the Delta-SCF method, we clearly resolve that the major deviation in the x-ray transition energies from Koopmans' theorem can be attributed to core-hole induced orbital relaxations.

Referee #2 (Remarks to the Author):

The manuscript by Z. Yin et al. reports the observation of photo-induced molecular dynamics in biology-related molecular dimers in solvent by laser-based soft-X-ray (SX) transient absorption spectroscopy (TAS). Experimentally, they used a table-top femtosecond SX source to measure transient absorption at carbon K edge up to a few picoseconds. The solvent target is prepared in the form of a flat liquid jet, which is very novel and original to the authors' group. Extensive quantum mechanical and molecular mechanics simulations allow to assign three bands at carbon K edge, and the observed shift of the peak position and absorption change are successfully attributed to proton transfer and molecular conformation respectively, with slightly different time scales.

Their discussions are highly convincing, illuminating the general features of intermolecular proton transfer and associated molecular structural changes. These observations cannot be done by conventional ultrafast spectroscopy with the visible light, and clearly showing the versatility of laser-based SX-TAS. This report will become a milestone in attosecond chemistry, which is emerging in the past several years after the development of attosecond technologies in the past two decades. I consider that this paper will meet the general scientific interests of the journal Nature.

We thank the referee for positive comments and the recommendation.

I thus recommend this manuscript to be published with minor revisions, provided that the authors answer to the following technical questions.

(1) In Fig. 3, the bands (i) and (ii) shows rapid decays, but the signals are revived after 1500 fs. These revivals are absent in the panel B. The authors should correct the panel B or explain this feature.

The feature in Figure 3a that the referee is referring to, appears to show a peak re-appearing at around 286.7 eV at late delays of about 1500 fs – 1700 fs. This was an artifact due to background subtraction that is now addressed in the revised manuscript.

The issue arises because the proton transfer band is superimposed on a shoulder of a broad peak at 289 eV as illustrated in Figure 2 (b).

We addressed the issue by extending the range of the spectral window that is used to fit the background slope as indicated in supplementary Figure S4. The new background subtraction includes more spectral points on the high energy side (extended by 1.3 eV) to better capture background contribution arising from the shoulder of the 289 eV peak.

Secondly, in the previously submitted manuscript, the choice of the colormap scale in Figure 3a was chosen such that some of the background points were saturated at the negative limit of the false color map scale which made it look like the signal of band (i) at 1500 fs – 1700 fs was re-appearing.

After this improvement of the background subtraction and adjustment of the false-color plot colormap scale, figure 3a has been updated in the revised manuscript.

(2) The authors claim that the oscillative features of Fig.2(E) is absent in the experimental data in Fig.2(A) because the scanning step is larger than the oscillation period. However, the sampling time is much shorter, and thus, the oscillation will be reflected in the experimental data as a fluctuation of the peak positions. This point should be clarified.

The referee is correct in pointing out that one might expect to observe a fluctuation of the line position in the experimental data. However, there are several reasons why such a fluctuation would be washed out in the experimental data. First, a 30-fs pump pulse was used in the experiment, whereas all ionization events were initiated simultaneously in the calculations. Taking the finite pump-pulse duration into account in the calculations would strongly reduce the oscillation amplitude. Second, the calculations shown in Fig. 2 are based on specific configurations in the liquid (namely urea dimer configurations), whereas the experimental data also reflects diffusion dynamics (as we discuss later in the text) that further washes out vibrational coherence. Finally, we note that the width of the experimentally observed “proton-transfer band” and the fluctuation width of its calculated equivalent are in good agreement, which further supports our assignment.

We have changed the following text (p. 7):

“They are not resolved in the experimental data owing to the larger delay-step size, **the ~30-fs instrument-response function and the additional contribution of diffusive dynamics in the experimental data.**”

(3) It is not clear how the experimental data shown in Fig. 3(B) corresponds to Fig. S9. At first thought, these two panels have a good correspondence, but the time scales are very different. Please clarify.

The band i in Fig. 3b as we state in the manuscript is coming from deeper hole states and can be compared to deeper hole state simulations, here compared to band i in Fig. 3f which comes from initially ionizing HOMO-3. This band i from the simulation can directly be compared to the timescale of nonadiabatic relaxation shown in Fig. S9a. The referee is correct to state that the timescale of Fig. S9 (now Fig S9a) is faster than the timescale of Fig. 3b. However, there can be even deeper ionization up to HOMO-5 as seen from Fig. S8. Relaxation time of HOMO-5 is also now shown in Fig. S9 along with HOMO-3 to highlight the fact that the relaxation timescale to HOMO is slower for this case. Additionally, we note that we only consider hydrogen bonded urea dimers in these simulations and these relaxation timescales for urea surrounded by water can be different.

(4) Regarding the experimental conditions, there is no explanation on the spot sizes for SX probe and visible pump beams. Since the pump process is highly nonlinear, I suppose that the pump spot is smaller than that of the probe SX. This point should be clarified.

We added the following description in the experimental setup of the supplementary information (Section 1.1.1):

“The soft X-ray spot size was estimated based on a pump drilled-pinhole to be ~ 62 μm through which the probe beam intensity was higher than 90 %. This is smaller than the 400 nm pump pulse with a spot size of ~ 80 μm . For details, see also the supplementary information of Ref. [2].”

Reviewer Reports on the First Revision:

Referee #1 (Remarks to the Author):

The authors addressed all concerns and the revised manuscript is excellent. I strongly recommend its publication. It will become an important milestone in the field.

Referee #2 (Remarks to the Author):

I read through the authors' reply, and I'm fully convinced by their comments. The corrections are all reasonable. It is indeed a beautiful work.